



# An integrated data compilation for the development of a marine protected area in the Weddell Sea

Katharina Teschke[1,2], Hendrik Pehlke[1,2], Volker Siegel[3], Horst Bornemann[1], Rainer Knust[1], Thomas Brey[1,2,4]

[1] Alfred Wegener Institute, Helmholtz Centre for Polar and Marine Research, Am Handelshafen 12, 27570 Bremerhaven, Germany
[2] Helmholtz Institute for Functional Marine Biodiversity at the University Oldenburg (HIFMB), Ammerländer Heerstraße 231, 26129 Oldenburg, Germany
[3] Thünen Institute of Sea Fisheries, Herwigstraße 31, 27572 Bremerhaven, Germany
[4] University Bremen, Bibliothekstraße 1, 28359 Bremen, Germany

*Correspondence to*: Katharina Teschke (Katharina.Teschke@awi.de)

Abstract. The Southern Ocean may contribute a considerable part to the proposed global network of Marine Protected Areas (MPAs) that should cover about 10% of the world oceans in 2020. In the Antarctic, the Commission for the Conservation of Antarctic Marine Living Resources (CCAMLR) is responsible for this task, and currently Germany leads a corresponding scientific evaluation of the wider Weddell Sea region. Compared to other marine regions within the Southern Ocean, the Weddell Sea is exceptionally well investigated. A tremendous amount of data and information has been produced over the last four decades. Here, we give a compilation of these data that were acquired in the context of the Weddell Sea MPA planning process. The data compilation comprises data produced by scientists / institutions from more than twenty countries and were either available within our institutes, provided by our collaborators, downloaded via data portals, or transcribed from the literature. It is the first data compilation for this area that includes abiotic data, such as bathymetry and sea ice, and ecological data from zooplankton, zoobenthos, fish, birds and marine mammals. The final data layer products based on this data compilation, including metadata description, are available from the data publisher PANGAE via the five persistent identifiers at https://doi.org/10.1594/PANGAEA.899520 (Pehlke et al., 2019a), https://doi.org/10.1594/PANGAEA.899591 (Teschke et al., 2019a), https://doi.org/10.1594/PANGAEA.899595 (Pehlke and Teschke, 2019), https://doi.org/10.1594/PANGAEA.899619 (Pehlke et al., 2019b), https://doi.org/10.1594/PANGAEA.899645 (Teschke et al., 2019b) and https://doi.org/10.1594/PANGAEA.899667 (Teschke et al., 2019c). This data compilation with the final data layer products will serve future research and monitoring well beyond the current MPA development process.

## 1 Introduction

Marine Protected Areas (MPAs) have experienced a significant increase in number and coverage at a global scale during recent decades (e.g. Mora and Sale, 2011; McDermott et al., 2018; UNEP-WCMC and IUCN, 2019). The number of MPAs has increased almost 1.5 times since the 1990s and the total area protected is currently almost 30 million km². At the United Nations World Summit on Sustainable Development in 2002 the international community of states reached an agreement about the establishment of a representative network of MPAs for the purposes of long-term conservation of marine biodiversity by 2012 (A/CONF.199/20, 2002). The adopted *strategic plan for biodiversity 2011-2020* of the Convention on Biological Diversity aims at the





conservation of at least 10 % of the coastal and offshore marine areas by 2020 based on a MPA network (CBD, 2010). The Southern Ocean may contribute a considerable proportion of this MPA network due to its size, and the uniqueness of the Antarctic environment renders its conservation the more urgent.

The Weddell Sea represents the southerly part of the Atlantic Sector of the Southern Ocean. About one quarter of the Weddell Sea's entire marine area covers the continental shelf along the eastern contour of the Antarctic Peninsula and the Antarctic continent up to 20°E as a non topographic delineation. The Weddell Sea is deserving protection in multiple respects. On the one hand, all arguments for the conservation of the Southern Ocean hold true for the Weddell Sea, too: An extreme environment mostly dominated by the seasonal dynamic of the sea ice with an excellent adapted biota. The biodiversity is - particularly in the benthos - very high (e.g. Brey et al.,

1994; Brandt et al., 2007), and there is a significant number of endemic species, i.e. unique to the Antarctic or even to the Weddell Sea (e.g. Arntz et al., 1994; Clarke and Johnston, 2003; Linse et al., 2006). Moreover, the Weddell Sea plays an important role for seabirds, penguins and marine mammals. Almost one third of the entire population of emperor penguins (Fretwell et al., 2012) and a major part of the circum-Antarctic population of crabeater seals (*cf.* Bester and Odendaal, 2000; Southwell et al., 2012; Gurarie et al. 2016) apparently occurs in

the Weddell Sea. Sponge associations which are comparable to tropical reef systems in terms of their structural and functional complexity occur along the eastern Weddell Sea shelf (Barthel and Gutt, 1992), and on the broad shelf in the southern Weddell Sea a special benthic community - adapted to very cold water temperatures - seems to resident (Teschke at al., 2016).

The Weddell Sea is - compared to other Antarctic regions - exceptionally well investigated. Since approximately

30 years the Weddell Sea is the geographical focus area of the German Antarctic research. In addition, there are manifold research activities of other nations. Consequently, we were able to compile a tremendous amount of environmental and ecological data to support the development of a Weddell Sea MPA (hereafter: WSMPA) under the Commission for the Conservation of Antarctic Marine Living Ressources (CCAMLR). Here we present a systematic overview of the integrated data compilation of environmental and ecological data collected

for the development of a WSMPA.

## 2  Data description

### 2.1 Study site

The WSMPA Planning Area in which we acquired the environmental and ecological data is located between the Antarctic Peninsula and 20°E (Fig. 1). The northern border is at 64°S and the continental margin forms the

southern border. This area is defined by CCAMLR's MPA Planning Domains (SC-CAMLR-XXX, 2011) and by aiming at a bio-geographically homogeneous area, particularly on the shelf (Teschke et al., 2016). In addition to the WSMPA Planning Area (approx. 4.2 million km² in size) we compiled data for a 200 km wide buffer area near the Antarctic Peninsula, which is part of an MPA initiative led by Argentina and Chile (CCAMLR-XXXVII/31, 2018). This buffer zone is adjacent northerly to the northern border of the WSMPA Planning Area

and has the eastern and western boundaries at 30°W and 60°W, respectively. Some data (e.g. seal tracking data), extend beyond the WSMPA Planning Area (plus buffer) and originate from adjacent regions of the Weddell Sea, such as the Bellinghausen Sea along the west side of the Antarctic Peninsula.



### 2.2 Data availability

All raw data sets of environmental and ecological parameters collected by the end of 2016 and further processed as part of the WSMPA planning process are systematically described and the primary reference is mentioned, such as the data portal on which the data are publically available, the institute/organisation on which the data can be requested on demand or the contact to the respective data provider (see Table 1 and 2; see all data records in Fig. 2 and Fig. S2). All raw data sets presented here were included in the WSMPA spatial planning analysis and the final data layer products with metadata description, including i.a. description of analytical data processing, are freely available from the data publisher PANGAEA via the five persistent identifiers at https://doi.org/10.1594/PANGAEA.899520 (Pehlke et al., 2019a), https://doi.org/10.1594/PANGAEA.899591 (Teschke et al., 2019a), https://doi.org/10.1594/PANGAEA.899595 (Pehlke and Teschke, 2019), https://doi.org/10.1594/PANGAEA.899619 (Pehlke et al., 2019b), https://doi.org/10.1594/PANGAEA.899645 (Teschke et al., 2019b) and https://doi.org/10.1594/PANGAEA.899667 (Teschke et al., 2019c) (see Table 1 and 2). The final layer products are available either as ArcMAP packages (as mxd file, containing a map document with all associated files) or as individual GIS files for those who use another GIS-software instead of the ESRI software (ArcMap). The shape and raster files were processed in such a way that they can be easily used for the analysis of MPA scenarios or other geostatistical analyses in the Weddell Sea.

### 2.3 Environmental data

All data sets on environmental parameters, which were collected in the context of the WSMPA planning, are mentioned here regardless of whether the environmental data have been used as basic data, e.g. in a regionalisation analysis of environmental provinces, or as explanatory variables for the distribution of a particular species (see Table 1).

#### 2.3.1 IBCSO data

The bathymetric data used in the context of the WSMPA planning initiative originate from the first regional digital bathymetric model (DBM) established in the International Bathymetric Chart of the Southern Ocean (IBCSO) Version 1.0 programme and published by Arndt et al. (2013a, b) (data request: April 2013) (Table 1; Fig. 3a). This chart model is based upon bathymetric data of different origin, such as multi-beam and single beam data, digitized depths from nautical charts, predicted bathymetry, from many hydrographic offices, scientific institutions and data centres. The IBCSO Version 1.0 DBM has a horizontal resolution of 500 m x 500 m and a vertical resolution of 1 m based on a polar stereographic projection with true scale at 65° referenced to WGS84 ellipsoid (Arndt et al., 2013b).

#### 2.3.2 AMSR-E sea ice maps

Daily high resolution sea ice maps of the Antarctic Ocean are provided by the PHAROS group (PHysical Analysis of RemOte Sensing images) at the Institute of Environmental Physics (IUP), University of Bremen, Germany. The sea ice raster maps, which were used in the context of the WSMPA planning initiative, are





derived from satellite observations of daily sea ice concentration by the Advanced Microwave Scanning Radiometer - Earth Observing System (AMSR-EOS) instrument on board the Aqua satellite. Daily AMSR-E sea ice concentration data (Jun 2002 - Oct 2011) were downloaded from IUP, University of Bremen (data request: 18-12-2013) (see Table 1; Fig. 3b). The ARTIST Sea Ice (ASI) concentration algorithm was used with a spatial

resolution of 6.25 km x 6.25 km (Spreen et al., 2008) and a polar stereographic projection (EPSG: 3976).

### 2.3.3 FESOM data

Monthly mean values of seawater temperature, salinity and current velocity from 1990 to 2009 were derived from the Finite Element Sea Ice - Ocean Model (FESOM) (Table 1; Fig. 3c, d). The model run was initialised on January, 1st 1980 with hydrographic data from the Polar Science Center Hydrographic Climatology (Steele et

al., 2001), and forced with NCEP daily atmospheric re-analysis data (Kalnay et al., 1996) for 1980 to 2009. For more information on FESOM and the atmospheric forcing data sets see e.g. Timmermann et al. (2009) and Haid and Timmermann (2013), respectively. The FESOM raster has a resolution of 0.18° (x) x 0.05° (y); in the vertical, two z-levels (i.e. sea surface and sea bottom) are used. The raster bases on WGS84 geographic coordinate system (EPSG: 4326).

### 2.3.4 SeaWiFS data

Near-surface chlorophyll a concentration values stem from the Sea-Viewing Wide Field-of-View Sensor (SeaWiFS) measurements on board of the OrbView-2 (formerly SeaStar) spacecraft (Table 1). The monthly aggregated data (1997 to 2010) were downloaded via the NASA's OceanColor website as level 3 standard mapped images (SMI) with a spatial resolution of 9 km x 9 km (data request: 09-09-2014).

### 2.3.5 WOA13 data

Data on dissolved oxygen, phosphate and nitrate were obtained from the World Ocean Atlas 2013 version 2 (WOA13 V2) (Garcia et al., 2014a, b) (Table 1). The data (1955 to 2012) were downloaded as monthly statistical means with a horizontal resolution of 1° (x) x 1° (y) and 57 and 37 vertical (z) levels between 0 to 1500 m and 0 to 500 m for dissolved oxygen and phosphate/nitrate, respectively. The data request was on 11-07-

2013 (dissolved oxygen), 17-07-2013 (nitrate) and 18-07-2013 (phosphate), respectively.

### 2.3.6 Data on chemical sediment components

A data compilation on total organic carbon content and calcium carbonate and silicia in surface sediments were downloaded from the data archive PANGAEA (Seiter et al., 2014a, b, c) (see Table 1). Data on biogenic silica of the sediment surface were obtained from PANGAEA, too (see Geibert et al., 2005a, b).



### 2.4 Ecological data

### 2.4.1 Zooplankton

The WSMPA data collection on adult Antarctic krill (*Euphausia superba*) originates from (i) historical UK data from "Discovery Expeditions" (1928-1939) and data collected during the SIBEX cruise by British Antarctic

Survey, (ii) five South African data sets from the 1990s, (iii) four Soviet data sets from 1998 and 1990, (iv) Polish data (Witek et al., 1985) and (v) German data from location discovery cruises with MV "Polarsirkel" in 1979/80 and 1980/81 (Siegel, 1982), RV "Walther Herwig" cruises (1975/76, 1977/78) and the 2004 Lazarev Sea Krill Survey (LAKRIS) (RV "Polarstern" cruises ANT-XXI/4) (Siegel, 2012). All the data are publicly available via the database KRILLBASE (doi.org/brg8) (Atkinson et al., 2017) (see Table S2 in the Supplement

that provides a detailed list of data used from KRILLBASE). The data from KRILLLBASE were complemented by abundance data on *E. superba*, which were collected *(a)* during the Norwegian Antarctic research expedition 1976/77 (MV "Polarsirkel") (Fevolden, 1979), *(b)* during two Soviet research cruises in 1977 (RV "Gizhiga") and 1983 (RV "Volny Vetter"), *(c)* in the context of the Lazarev Sea Krill Survey (RV "Polarstern" cruises ANT-XXIII/2, ANT-XXIII/6, ANT-XXIV/2) (e.g. Siegel, 2012) as well as (d) during RV "Polarstern" cruise

ANT-V/1-3, ANT-VII/4, ANT-XVIII/4 and ANT-XXIX/3 (Siegel et al., 2013) (Table 2). Furthermore, Japanese, Norwegian and Soviet fisheries data (catch and effort) on *E. superba* for the WSMPA Planning Area (Statistical Subarea 48.5 and southern part of Subarea 48.6) stem from the CCAMLR database (data request through CCAMLR Secretariat: 03-10-2013) (Table 2).

Abundance data on Antarctic krill larvae stem from (a) the Antarctic research expeditions 1976/77 (Fevolden,

1979) and 1979/80 with MV "Polarsirkel" (Siegel, 1982), (b) the First International BIOMASS Experiment survey (FIBEX) (RV "Walther Herwig" cruise 1981) (e.g. Trathan and Everson, 1994) and the Lazarev Sea Krill Survey (LAKRIS) (RV "Polarstern" cruises ANT-XXI/4, ANT-XXIII/6) (Siegel, 2012) as well as (c) RV "Polarstern" cruise ANT-VII/4 and the combined RV "Polarstern" (ANT-VIII/2) and RV "Akademik Fedorov" cruise (Menshenina, 1992) (see Table 2).

Abundance data on adult ice krill (*Euphausia crystallorophias*) originate from pelagic trawl surveys during (a) the German Antarctic research cruise 1975/76 with "Walther Herwig", (b) the "Pre-Site Survey" 1979/80 with MV "Polarsirkel" (Siegel, 1982), (c) the Lazarev Sea Krill Survey (RV "Polarstern" cruises ANT-XXI/4, ANT-XXIII/2, ANT-XXIII/6, ANT-XXIV/2) (e.g. Siegel, 2012) as well as (d) RV "Polarstern" cruise ANT-V/1-3, ANT-VII/4 and ANT-XXIX/3 (Siegel et al., 2013) (Table 2).

All data about *E. superba* and *E. crystallorophias,* which were used additionally to KRILLBASE and the CCAMLR database, are stored in the data warehouse of the Thuenen Institute of Sea Fisheries (https://www.thuenen.de) and can be requested on demand.

### 2.4.2 Zoobenthos

Abundance data and presence-absence records on sponges (higher taxonomic groups), which were compiled in

the context of the WSMPA planning initiative, originate from zoobenthos data sets, which are publically available via PANGAEA (see Gerdes, 2014 a-o; Teschke and Brey, 2019a) (see Table 2). The data set on echinoderms consists of presence-absence data on species level for asteroids, abundance data on ophiuroid taxa as well as holothurian taxa. The first two data sets are available in PANGAEA (Teschke and Brey, 2019b, c), the





latter in the information system biodiversity.aq (Gutt et al., 2014). Publications, which have used those primary data sets, are e.g. Dahm (1996), Gutt (1988) and Gerdes et al. (1992).

### 2.4.3 Fish

The WSMPA data collection on Antarctic silverfish larvae (*Pleuragramma antarctica*) originates from
quantitative zooplankton data sets obtained during the "Polarstern" cruises ANT-I/2 (Boysen-Ennen and Piatkowski, 1988) and ANT-III/3 (Hubold et al., 1988) and during the Lazarev Sea Krill Survey (LAKRIS) ("Polarstern" cruises: ANT-XXI/4, ANT-XXIII/6, ANT-XXIV/2) (Flores et al., 2014) (Table 2). The first mentioned data are stored in the data warehouse of the Thuenen Institute of Sea Fisheries and can be requested on demand. The LAKRIS data are available in PANGAEA (*PANGAEA reference will be added during review*
*process*).

Abundance data on demersal fish and adult *P. antarctica* stem from benthic and pelagic trawl surveys during seven "Polarstern" cruises between 1996 and 2011 (ANT-XIII/3, ANT-XV/3, ANT-XVII/3, ANT-XIX/5, ANT-XXI/2, ANT-XXIII/8, ANT-XXVII/3) (Table 2). Publications, which have used these data, are e.g. Caccavo et al. (2018) and Mintenbeck et al. (2012). The primary datasets can be requested from us if required (contact:
Rainer Knust, AWI) and will be available in PANGAEA without any restrictions by autumn 2019.  This data compilation was complemented by data on demersal fish and *P. antarctica* derived from trawl and dredge surveys published in PANGAEA (Drescher et., 2012; Ekau et al., 2012a, b; Hureau et al., 2012; Kock et al., 2012; Wöhrmann et al., 2012).

Fishery data (catch per unit effort) on the Antarctic toothfish (*Dissostichus mawsoni*) for the WSMPA Planning
Area (Statistical Subarea 48.5 and southern part of Subarea 48.6) were taken from the CCAMLR database and requested through the CCAMLR Secretariat (data request: 03-08-2016) (Table 2).

Information about nesting sites of demersal fish was collected during the RV "Polarstern" cruises PS82 (ANT-XXIX/9) and PS96 (ANT-XXXI/2). The data are available from Knust and Schröder (2014) (PS82) and Piepenburg (2016) (PS96). The data collected during RV "Polarstern" cruises were supplemented by data from
the literature (Daniels 1978, 1979; Jones & Near 2012).

### 2.4.4 Flying and non-flying seabirds

Tracking data on breeding Adélie penguins *(Pygoscelis adeliae)* originate from (i) British Antarctic Survey (BAS) inventory data from Phil Trathan (ID 754) and Mike Dunn and P. Trathan (ID 764, 773, 779), (ii) a data
set from BAS (P. Trathan) and Instituto Antártico Argentino (Mercedes Santos) (ID 753) (Warwick et al., 2019) and (iii) a data set from the US AMLR Program from Jefferson Hinke and Wayne Trivelpiece (NOAA) (ID 910) (see e.g. Hinke et al. 2015) (Table 2). All the data are stored in the Birdlife International`s Seabird Tracking Database (data request: 20-10-2015). Adélie penguins breeding locations and estimated abundances of breeding pairs were derived from Lynch and LaRue (2014).

Tracking data on non-breeding *P. adeliae* were acquired from Birdlife International`s Seabird Tracking Database, too (data request: 20-10-2015) (Table 2). Downloaded data include (i) BAS inventory data from Phil Trathan (ID 754) and Mike Dunn and P. Trathan (ID 773, 779), (ii) a data set from BAS (P. Trathan) and





Instituto Antártico Argentino (Mercedes Santos) (ID 753) and (iii) a data set from the US AMLR Program from Jefferson Hinke and Wayne Trivelpiece (NOAA) (ID 910).

Data on Emperor penguin *(Aptenodytes forsteri)* colony locations and breeding population estimates were derived from Fretwell et al. (2012, 2014) (Table 2).

Information on breeding locations and estimated number of breeding pairs of the Antarctic petrel *(Thalassoica antarctica)* were kindly provided by Jan van Franeker (Wageningen University & Research) and are published in van Franeker et al. (1999) (Table 2).

### 2.4.5 Pinnipeds

Tracking data from pinnipeds were obtained from the MEOP data portal "Marine Mammals Exploring the
Oceans Pole to Pole" (data request: 14-11-2016) (see Table 2 for a detailed list of data used). In addition, we have used MEOP data (UK data: ct27, ct70; German data: ct113, wd06, wd07) for which unconditional sharing were not yet accepted at the time of data retrieval and were provided by Lars Boehme (University of St. Andrews) and us (H. Bornemann), respectively. The UK and German data sets are now also freely accessible from the MEOP data portal.

Furthermore, the data from the MEOP data portal were complemented by tracking data sets on southern elephant seals (Tosh et al., 2009a, b; James et al., 2012a, b), Weddell seals (McIntyre et al., 2013a, b) and crabeater seals (Nachtsheim et al., 2016a, b) stored in PANGAEA.

Point data from pack-ice seals (unspecified taxa) based on aerial surveys are from Plötz et al. (2011a-e) (Table 2). These data were sampled during five flight campaigns from 1996 to 2001 within the Antarctic Pack Ice Seals
(APIS) programme. Data were downloaded from PANGAEA. In addition, point data on pack-ice seals from the South African APIS census programme were downloaded from PANGAEA (Bester and Odendaal, 2015), and the data from the UK APIS census programme are available from either Phil Trathan (BAS), or the NERC UK Polar Data Centre (Table 2). German and South African APIS data and UK census data were published in e.g. Gurarie et al. (2016) and Forcada et al. (2012), respectively.

### 3 Outlook

This is the first data compilation for the Antarctic Weddell Sea and adjacent seas, which considers data across the entire ecosystem: i.e., from abiotic data, such as bathymetry and sea ice, to ecological data ranging from zooplankton and zoobenthos to fish, birds and marine mammals. The effort to create such a data compilation was directly coupled with the initiative to develop a WSMPA. However, the data compilation is also suitable for
further scientific questions in the wide field of faunistic, ecological and nature conservation studies to investigate the effect of climate change and possible fishing activities in this area. Some of the ecological data sets were collected in the 1980s and earlier, when the Weddell Sea was still almost pristine and hardly affected by any anthropogenic activities, so that these data sets are optimally suited to describe a reference state for assessing the effect of pressures on the Weddell Sea ecosystem. In addition, the ecological data - with a few exceptions -
provide information on abundances of the respective taxa and are therefore better suited as an indicator for environmental changes than presence-absence data or presence data only.



Ultimately, the data compilation serves to protect our data heritage for use by future generations (baseline is a particular issue), to enable work with readily available multi-parameter data sets, and to motivate researches to add further data, both from existing "paper sources" and from future measurements.

Subsequent work will focus on the development of an efficient and tailor-made management system for the storage of these complex and heterogeneous data and information of WSMPA data compilation and automated data mining, handling and analysis. This system will serve three purposes: (i) to better enable a more holistic and integrative approach towards ecosystem research in the Weddell Sea in general, (ii) to enable the management of the WSMPA to carry out the tasks of the Research and Monitoring Programme as a mandatory part of an MPA under CCAMLR when adopting the MPA, and (iii) to provide key stakeholders and the public with access to

data, information and management measures related to the ecosystem of the Weddell Sea region in general and the WSMPA in particular.

**Author contribution.** KT collected all data together, described the metadata and led the writing of the paper. HP took over the technical part of the data acquisition (retrieval, storage, processing). VS collected and prepared the data on zooplankton for further analyses within the WSMPA planning. HB and RK were significantly involved

in the collection of the data on pinnipeds and fishes, respectively. TB collaborated in the paper writing.

**Competing interests.** The authors declare that they have no conflict of interests.

**Acknowledgements.** The project was financially supported by the German Federal Ministry of Food and Agriculture. In particular, we would like to thank all colleagues from all national and international scientific institutions who have supported us in providing data used to build up the data compilation for the wider Weddell

Sea: i.e. from the Alfred Wegener Institute (Hauke Flores, Dieter Gerdes, Julian Gutt, Stefan Hain, Kerstin Jerosch, Rainer Knust, Dieter Piepenburg, Ralf Timmermann), British Antarctic Survey (Phil Trathan), CCAMLR Secretariat (Elanor Miller, Tim Jones, David Ramm), Helmholtz Centre Geesthacht (Verena Haid), Instituto Antártico Argentino (Mercedes Santos), National Oceanic and Atmospheric Administration (Jefferson Hinke), Royal Belgian Institute of Natural Sciences (Anton van de Putte), Stony Brook University (Heather

Lynch), Thünen Institute of Sea Fisheries (Karl-Hermann Kock), University of Gothenburg (Tomas Lundälv), University of Padova (Emilio Riginella), University of St. Andrews (Lars Boehme), Wageningen University & Research (Jan van Franeker). The marine mammal data were collected and made freely available by the International MEOP Consortium and the national programs that contribute to it (http://www.meop.net). The seal tracking data ct96 and ct109 are collected by the Integrated Marine Observing System (IMOS). IMOS is a

national collaborative research infrastructure, supported by the Australian Government. It is operated by a consortium of institutions as an unincorporated joint venture, with the University of Tasmania as Lead Agent.
**Figure 1.**

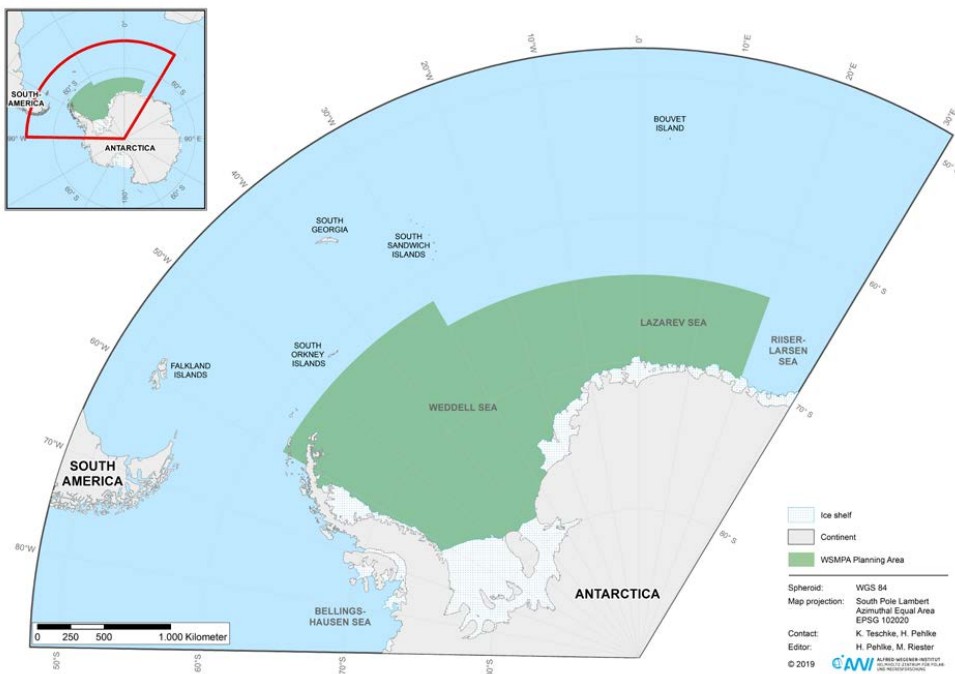

5    **Figure 1.** Study site in the Antarctic Weddell Sea and adjacent marine regions. Black dashed line indicates the boundaries of Weddell Sea MPA Planning Area including the 200 km wide buffer area near the Antarctic Peninsula. Overview map of the study site in the wider Weddell Sea and its location in the Southern Ocean (top left corner).

**Figure 2.**

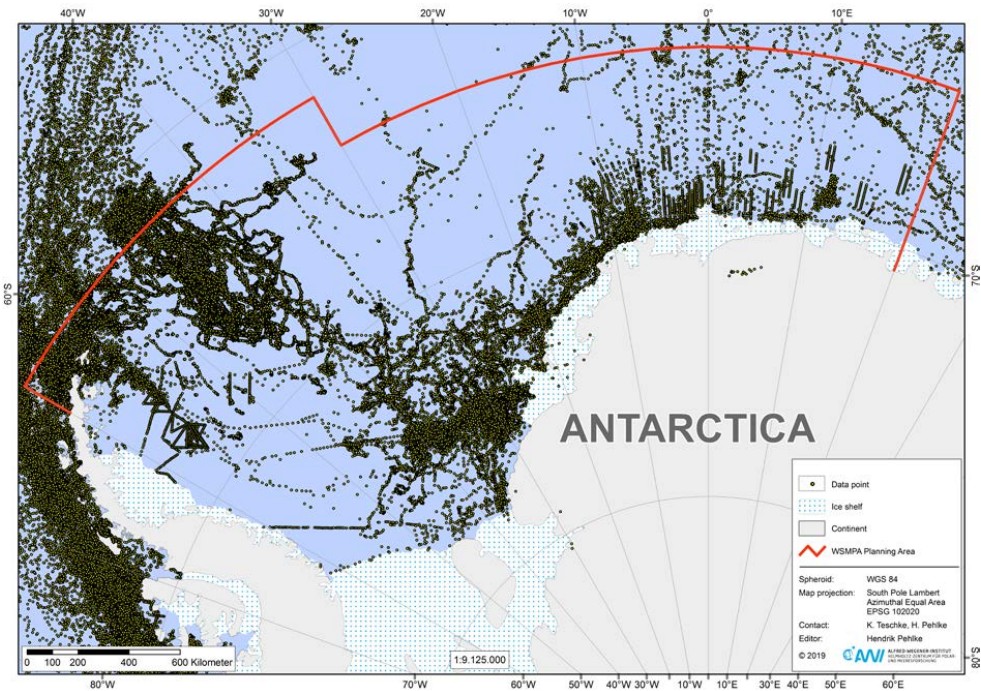

5    **Figure 2.** Distribution of all data recordings across the wider Weddell Sea region, which were compiled in the context of the WSMPA planning initiative. Figure S2 in the Supplement provides the distribution of data recordings per higher taxonomic group, i.e. zooplankton, zoobenthos, fishes, birds and pinnipeds.

**Figure 3.**

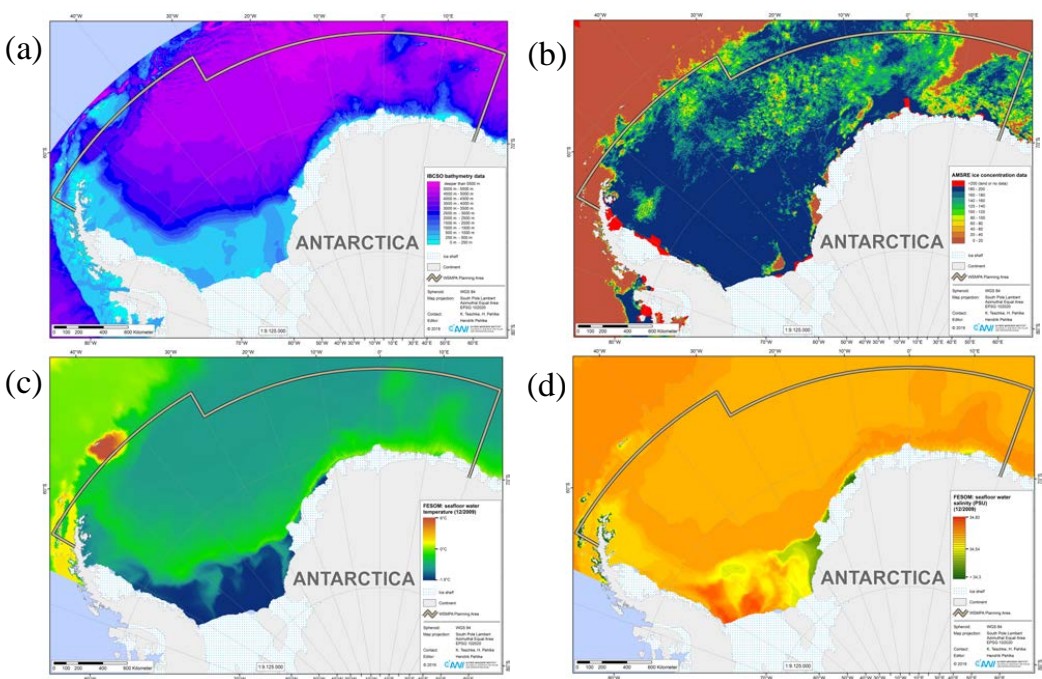

**Figure 3.** Raster data sets of environmental parameters, which have been used as basic data in a regionalisation analysis of environmental provinces in the context of the WSMPA planning. IBCSO bathymetry (a), AMSR-E sea ice maps (exemplarily for 15 December 2009) (b), FESOM temperature and salinity data (exemplarily for December 2009) (c, d).



**Tables**

**Table 1.** Data collection of environmental parameters compiled for the development of a marine protected area (MPA) in the wider Weddell Sea (Antarctica). For each raw data set, the name of the data source, the primary reference, such as the data portal on which the data is publicly accessible or the contact to the respective data provider, as well as examples of publications that have used the respective primary data set are listed. In addition, DOI links to the final WSMPA data layer products is provided, which includes the respective raw data set.

| Content of Data | Name of data source | Reference to primary data set or data provider | Reference to publications, which have used primary data set (exemplarily) | DOI link to ArcMap packages |
|---|---|---|---|---|
| Depth | International Bathymetric Chart of the Southern Ocean (IBCSO) Version 1.0 | Arndt et al. (2013a) [data request: April 2013] | Arndt et al. (2013b), Jerosch et al. (2016) | https://doi.pangaea.de/10.1594/PANGAEA.899595, https://doi.pangaea.de/10.1594/PANGAEA.899667, https://doi.pangaea.de/10.1594/PANGAEA.899591 |
| Temperature, salinity, current velocity | Finite Element Sea Ice – Ocean Model (FESOM) | Contact: Ralph Timmermann (AWI) ralph.timmermann@awi.de [data delivery: 20-11-2013] | Danilov et al. (2004), Timmermann et al. (2009), Haid and Timmerman (2013) | https://doi.pangaea.de/10.1594/PANGAEA.899595, https://doi.pangaea.de/10.1594/PANGAEA.899645, https://doi.pangaea.de/10.1594/PANGAEA.899667, https://doi.pangaea.de/10.1594/PANGAEA.899591 |
| Sea ice concentration | Daily AMSR-E Sea Ice Maps | https://seaice.uni-bremen.de/data/ Contact: Gunnar Spreen, Christian Melsheimer or Georg Heygster (Institute of Environmental Physics, University of Bremen) [data request: 18-12-2013] | Spreen et al. (2008) | https://doi.pangaea.de/10.1594/PANGAEA.899595, https://doi.pangaea.de/10.1594/PANGAEA.899667 |
| Chlorophyll a concentration | Sea-Viewing Wide Field-of-View Sensor (SeaWiFS) measurements | NASA's OceanColor website [data request: 09-09-2014] | Moore and Abbott (2000), Gregg and Casey (2004) | https://doi.pangaea.de/10.1594/PANGAEA.899667 |
| Dissolved oxygen, phosphate, nitrate | World Ocean Atlas 2013 version 2 (WOA13 V2) | https://www.nodc.noaa.gov/OC5/woa13/woa13data.html [data request: 11 to 18 July 2013] | Garcia et al. (2014a, b) | https://doi.pangaea.de/10.1594/PANGAEA.899667, https://doi.pangaea.de/10.1594/PANGAEA.899591 |
| Total organic carbon content | | Seiter et al. (2014b) | Seiter et al. (2014a) | https://doi.pangaea.de/10.1594/PANGAEA.899591 |
| Calcium carbonate, silica | | Seiter et al. (2014c) | Seiter et al. (2014a) | https://doi.pangaea.de/10.1594/PANGAEA.899591 |
| Biogenic silica | | Geibert et al. (2005b) | Geibert et al. (2005a) | https://doi.pangaea.de/10.1594/PANGAEA.899591 |



**Table 2.** Data collection of ecological parameters compiled for the development of a marine protected area (MPA) in the wider Weddell Sea (Antarctica). For each raw data set, the name of the data source, the primary reference, such as the data portal on which the data is publicly accessible or the contact to the respective data provider, as well as the respective cruise reports and/or examples of publications that have used the respective primary data set are listed. In addition, DOI links to the final WSMPA data layer products is provided, which includes the respective raw data set. The raw data sets are structured according to higher taxonomic groups, i.e. zooplankton, zoobenthos, fishes, birds and pinnipeds.

Within each higher taxonomic group, the individual raw data sets are sorted by taxa and the oldest data set is listed first each time.

| Data content | Name of data source | Reference to primary data set or data provider | Cruise reports | Reference to publications, which have used primary data set (exemplarily) | DOI link to ArcMap packages |
|---|---|---|---|---|---|
| **Zooplankton** | | | | | |
| Adult Antarctic krill (abundances) | KRILLBASE (doi.org/brg8) See detailed list of data in Table S2 in Supplement | Atkinson et al. (2017) | | Atkinson et al. (2004) Atkinson et al. (2008) Piñones and Fedorov (2016) Atkinson et al. (2019) | https://doi.pangaea.de/10.1594/PANGAEA.899667 |
| Adult Antarctic krill (catch and effort) | Japanese, Norwegian and Soviet fisheries data | CCAMLR database; Contact: CCAMLR Secretariat [data request: 03-10-2013] | | | https://doi.pangaea.de/10.1594/PANGAEA.899667 |
| Adult Antarctic krill (abundances) | MV Polarsirkel 1976/77 | Database of Thuenen Institute of Sea Fisheries | | Fevolden (1979) | https://doi.pangaea.de/10.1594/PANGAEA.899667 |
| Adult Antarctic krill (abundances) | Soviet cruises: RV Gizhiga 1977 and RV Volny Vetter 1983 | Database of Thuenen Institute of Sea Fisheries | | | https://doi.pangaea.de/10.1594/PANGAEA.899667 |
| Adult Antarctic krill (abundances) | ANT-XVIII/4 | Database of Thuenen Institute of Sea Fisheries | Fahrbach et al. (2003) | | https://doi.pangaea.de/10.1594/PANGAEA.899667 |
| Adult Antarctic krill & ice (abundances) | Lazarev Sea Krill Survey (LAKRIS) data (ANT-XXI/4, ANT-XXIII/2, ANT-XXIII/6, ANT-XXIV/2) | Database of Thuenen Institute of Sea Fisheries | Smetacek et al. (2005) Strass (2007) Bathmann (2008, 2010) | Siegel (2012) | https://doi.pangaea.de/10.1594/PANGAEA.899667 |
| Adult Antarctic krill & ice (abundances) | ANT-XXIX/3 | Database of Thuenen Institute of Sea Fisheries | Gutt (2013) | Siegel et al. (2013) | https://doi.pangaea.de/10.1594/PANGAEA.899667 |
| Adult Antarctic krill & ice (abundances) | ANT-V/3 | Database of Thuenen Institute of Sea Fisheries | Schnack-Schiel (1987) | | https://doi.pangaea.de/10.1594/PANGAEA.899667 |
| Adult Antarctic krill & ice (abundances) | ANT-VII/4 | Database of Thuenen Institute of Sea Fisheries | Arntz et al. (1990) | | https://doi.pangaea.de/10.1594/PANGAEA.899667 |





| Data type | Expedition / Survey | Source | Reference (a) | Reference (b) | DOI |
|---|---|---|---|---|---|
| Adult ice krill (abundances) | RV Walther Herwig 1975/76 | Database of Thuenen Institute of Sea Fisheries | | | https://doi.pangaea.de/10.1594/PANGAEA.899667 |
| Adult ice krill (abundances) | MV Polarsirkel 1979/80 | Database of Thuenen Institute of Sea Fisheries | Siegel (1982) Hempel et al. (1983) | | https://doi.pangaea.de/10.1594/PANGAEA.899667 |
| Larval Antarctic krill (abundances) | MV Polarsirkel 1976/77 MV Polarsirkel 1979/80 | Database of Thuenen Institute of Sea Fisheries | Fevolden (1979) Siegel (1982) | | https://doi.pangaea.de/10.1594/PANGAEA.899667 |
| Larval Antarctic krill (abundances) | First International BIOMASS Experiment survey (FIBEX), RV "Walther Herwig" 1981 | Database of Thuenen Institute of Sea Fisheries | Trathan and Everson (1994) Siegel (2005) | | https://doi.pangaea.de/10.1594/PANGAEA.899667 |
| Larval Antarctic krill (abundances) | ANT-VII/4 | Database of Thuenen Institute of Sea Fisheries | | Arntz et al. (1990) | https://doi.pangaea.de/10.1594/PANGAEA.899667 |
| Larval Antarctic krill (abundances) | ANT-VIII/2 and RV Akademik Fedorov, 1989 | Database of Thuenen Institute of Sea Fisheries | Menshenina (1992) | Augstein et al. (1991) | https://doi.pangaea.de/10.1594/PANGAEA.899667 |
| Larval Antarctic krill (abundances) | Lazarev Sea Krill Survey (LAKRIS) data (ANT-XXI/4, ANT-XXIII/6) | Database of Thuenen Institute of Sea Fisheries | Siegel (2012) | Smetacek et al. (2005) Bathmann (2008) | https://doi.pangaea.de/10.1594/PANGAEA.899667 |

**Zoobenthos**

| Data type | Expedition / Survey | Source | Reference (a) | Reference (b) | DOI |
|---|---|---|---|---|---|
| Sponges (abundances) | ANT-III/2, WH85 ANT-V/1 ANT-VI/3 ANT-VII/4 ANT-IX/3 ANT-X/3 ANT-XIII/3 ANT-XIII/4 ANT-XV/3 ANT-XVII/3 ANT-XIX/5 ANT-XXI/2 ANT-XXIII ANT-XXVII | Gerdes (2014a-o) | Gerdes et al. (1992) Gerdes et al. (2003) | Hempel (1985) Schnack-Schiel (1987) Fütterer (1988) Arntz et al. (1990) Bathmann et al. (1992) Spindler et al. (1993) Arntz and Gutt (1997) Fahrbach and Gerdes (1997) Arntz and Gutt (1999) Arntz and Brey (2001) Arntz and Brey (2003) Arntz and Brey (2005) Gutt (2008) Knust et al. (2012) | https://doi.pangaea.de/10.1594/PANGAEA.899645 |



| Taxon | Cruise | References | Data citation | Further references | DOI |
|---|---|---|---|---|---|
| Sponges (presence-absence) | ANT-VII/4, ANT-IX/3, ANT-XIII/3, ANT-XV/3, ANT-XVII/3, ANT-XXI/2 | Arntz et al. (1990), Bathmann et al. (1992), Arntz and Gutt (1997), Arntz and Gutt (1999), Arntz and Brey (2001), Arntz and Brey (2005) | Teschke and Brey (2019a) | Galéron et al. (1992) | https://doi.pangaea.de/10.1594/PANGAEA.899645 |
| Echinoderms - Asteroids (presence-absence) | ANT-I/2, ANT-II/4, ANT-V/3, ANT-VI/3, ANT-XV/3, ANT-XVII/3 | Drescher et al. (1983), Kohnen (1984), Schnack-Schiel (1987), Fütterer (1988), Arntz and Gutt (1999), Arntz and Brey (2001) | Teschke and Brey (2019b) | Voß (1988) | https://doi.pangaea.de/10.1594/PANGAEA.899645 |
| Echinoderms - Ophiuroids (abundances) | ANT-I/2, ANT-II/4, ANT-V/3, ANT-V/4, ANT-VI/3, ANT-VII/3, ANT-VII/4, ANT-IX/3, ANT-X/3 | Drescher et al. (1983), Kohnen (1984), Schnack-Schiel (1987), Miller and Oerter (1990), Fütterer (1988), Arntz et al. (1990), Bathmann et al. (1992), Spindler et al. (1993) | Teschke and Brey (2019c) | Brey et al. (1994), Dahm (1996) | https://doi.pangaea.de/10.1594/PANGAEA.899645 |
| Echinoderms - Holothurians (abundances) | ANT-I/2, ANT-II/4, ANT-III/3 | Drescher et al. (1983), Kohnen (1984), Hempel (1985) | Gutt, Piepenburg and Voß (2014) | Gutt (1988), Piepenburg et al. (1997) | https://doi.pangaea.de/10.1594/PANGAEA.899645 |
| **Fishes** | | | | | |
| Fish larvae - *Pleuragramma antarctica* (abundances) | ANT-I/2 | Drescher et al (1983) | Database of Thuenen Institute of Sea Fisheries | Boysen-Ennen and Piatkowski (1988) | https://doi.pangaea.de/10.1594/PANGAEA.899591 |
| Fish larvae - *Pleuragramma antarctica* (abundances) | ANT-III/3 | Hempel (1985) | Hubold et al. (1988) | | https://doi.pangaea.de/10.1594/PANGAEA.899591 |
| Fish larvae - *Pleuragramma antarctica* (abundances) | Lazarev Sea Krill Survey (LAKRIS) data: ANT-XXI/4, ANT-XXIII/6, ANT-XXIV/2 | Smetacek et al. (2005), Bathmann (2008, 2010) | *PANGAEA reference will be added during review process* | Flores et al. (2014) | https://doi.pangaea.de/10.1594/PANGAEA.899591 |



| Data | Dataset / Source | Data reference | Publication reference | URL |
|---|---|---|---|---|
| *Pleurogramma antarctica* & demersal fishes (abundances) | ANT-XIII/3, ANT-XV/3, ANT-XVII/3, ANT-XIX/5, ANT-XXI/2, ANT-XXVII/3 | Contact: Rainer Knust (AWI) Rainer.Knust@awi.de | Arntz and Gutt (1997), Arntz and Gutt (1999), Arntz and Brey (2001), Arntz and Brey (2003), Arntz and Brey (2005), Knust et al. (2012) — Mintenbeck et al. (2012), Caccavo et al. (2018) | https://doi.pangaea.de/10.1594/PANGAEA.899591 |
| *Pleurogramma antarctica* & demersal fishes (abundances) | ANT-I/2, ANT-III/3, ANT-V/3, ANT-VII/4, ANT-IX/3, ANT-XXIII/8 | Drescher et al. (2012), Ekau et al. (2012a), Ekau et al. (2012b), Hureau et al. (2012), Wöhrmann et al. (2012), Kock et al. (2012) | Drescher et al (1983), Hempel (1985), Schnack-Schiel (1987), Arntz et al. (1990), Bathmann et al. (1992), Gutt (2008) — Ekau (1988), Caccavo et al. (2018) | https://doi.pangaea.de/10.1594/PANGAEA.899591 |
| Antarctic toothfish (catch per unit effort) | Japanese, Korean, Norwegian and South African fishing data | CCAMLR database; Contact: CCAMLR Secretariat [data request: 03-08-2016] | | https://doi.pangaea.de/10.1594/PANGAEA.899591 |
| Demersal fish nesting sites | PS82 (ANT-XXIX/9), PS96 (ANT-XXXI/2) | Knust and Schröder (2014), Piepenburg (2016) | Schröder (2016) | https://doi.pangaea.de/10.1594/PANGAEA.899591 |
| Demersal fish nesting sites | | Daniels (1978, 1979), Jones and Near (2012) | La Mesa et al. (2019) | https://doi.pangaea.de/10.1594/PANGAEA.899591 |

**Birds**

| Data | Dataset / Source | Data reference | Publication reference | URL |
|---|---|---|---|---|
| Adélie penguin colonies (estimated abundances of breeding pairs) | | Lynch and LaRue (2014) | | https://doi.pangaea.de/10.1594/PANGAEA.899520 |
| Breeding and non-breeding Adélie penguins (tracking data) | US AMLR Program (ID 910) | Birdlife International`s Seabird Tracking Database [data request: 20-10-2015] | Hinke et al. (2015) | https://doi.pangaea.de/10.1594/PANGAEA.899520 |
| Breeding and non-breeding Adélie penguins (tracking data) | BAS / Instituto Antártico Argentino data (ID 753) | Birdlife International`s Seabird Tracking Database [data request: 20-10-2015] | Warwick-Evans et al. (2019) | https://doi.pangaea.de/10.1594/PANGAEA.899520 |
| Breeding and non-breeding Adélie penguins (tracking data) | BAS Inventory (754, 773, 779) | Birdlife International`s Seabird Tracking Database [data request: 20-10-2015] | Dunn et al. (2011) | https://doi.pangaea.de/10.1594/PANGAEA.899520 |
| Breeding Adélie penguins (tracking data) | BAS Inventory (ID 764) | Birdlife International`s Seabird Tracking Database [data request: 20-10-2015] | Lynnes et al. (2002) | https://doi.pangaea.de/10.1594/PANGAEA.899520 |
| Emperor penguin colonies (populations estimates) | | Fretwell et al. (2012), Fretwell et al. (2014) | | https://doi.pangaea.de/10.1594/PANGAEA.899520 |



| Taxa | Data source / notes | Citation | Link |
|---|---|---|---|
| Antarctic petrel Colonies (estimated number of breeding pairs) | | Van Franeker et al. (1999) | https://doi.pangaea.de/10.1594/PANGAEA.899520 |
| **Pinnipeds** | | | |
| Seal taxa (tracking data) | Data from: Australia (ct109, ct96), Brazil (ct56, ct46, ct39, ct22), China (ct105), UK (ct1, ct8, ct27, ct27x, ct40, ct43, ct45, ct49, ct58, ct70), France (ct16, ct62, ft01, ft02, ft11, ft12), Germany (ct21, ct35, ct35b, ct44, ct54, ct68, ct87, ct99, ct102, ct113, wd06, wd07), Norway (ct34), South Africa (ct33, ct50, ct73), USA (ct9, ct14, ct25, ct29, ct37, ct48) | Treasure et al. (2017) Nachtsheim et al. (2019) Boehme et al. (2016) Marine Mammals Exploring the Oceans Pole to Pole (MEOP) data portal [data request: 14-11-2016] | https://doi.pangaea.de/10.1594/PANGAEA.899619 |
| Southern elephant seals (tracking data) | Tosh et al. (2009a) James et al. (2012a) | Tosh et al. (2009b) James et al. (2012b) | https://doi.pangaea.de/10.1594/PANGAEA.899619 |
| Weddell seals (tracking data) | McIntyre et al. (2013a) | McIntyre et al. (2013b) | https://doi.pangaea.de/10.1594/PANGAEA.899619 |
| Crabeater seals (tracking data) | Nachtsheim et al. (2016a) | Nachtsheim et al. (2016b) | https://doi.pangaea.de/10.1594/PANGAEA.899619 |
| Pack-ice seals (aerial surveys) | Plötz et al. (2011a-e) Antarctic Pack Ice Seals (APIS) programme EMAGE-I to -V | Southwell et al. (2012) Gurarie et al. (2016) | https://doi.pangaea.de/10.1594/PANGAEA.899619 |
| Pack-ice seals (aerial surveys) | Bester and Odendaal (2015) South African APIS census | Arntz and Gutt (1999) Southwell et al. (2012) Gurarie et al. (2016) | https://doi.pangaea.de/10.1594/PANGAEA.899619 |
| Pack-ice seals (aerial surveys) | NERC UK Polar Data Centre, polardatacentre@bas.ac.uk; Phil Trathan (BAS), pnt@bas.ac.uk [data delivery: 16-10-2013] UK APIS census | Forcada et al. (2012) | https://doi.pangaea.de/10.1594/PANGAEA.899619 |



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
