# Peer review of "An integrated data compilation for the development of a marine protected area in the Weddell Sea"

_Earth System Science Data, 2019_

## Referee Comment (RC1) · Anonymous Referee #1 · 6 Nov 2019

General comments The authors provide data layer products that are based on a huge data compilation of environmental and ecological data from the Weddell Sea, used in the planning process for a Weddell Sea MPA. The data layer products can be easily accessed and downloaded from the data portal PANGAEA via provided persistent identifiers. The data format (either ESRI ArcMap packages or separate shapefiles) is common, I used ArcMap 10.3.1 to view the files. The data compilation itself is not directly accessible, the sources to all used raw data are thoroughly listed in part 2 of the paper and in Table 1 and 2. The data layer products themselves are not really described in the paper. The methods used to create the shapefiles from the compiled raw data are rather briefly stated on the respective PANGAEA sites. For further information

on the spatial analysis a link to CCAMLR working papers is provided. Following this link, however, leads to the CCAMLR webpage where for each report a formal request has to be submitted. I did so for one report and (so far) did not receive the document. How or if the provided data layer products might be used, without access to the underlying data compilation, is not discussed.

Specific comments At first reading of title and abstract I was under the impression that a huge data compilation is provided with this paper, while in fact a systematic overview of all the sources to the MPA planning process is given. This should be made more clear right from the begin, so the reader knows what to expect. Usability of data: The authors should elaborate if and how the provided data and information might be used, and/or to which future work they might contribute. Can the provided data layer products directly be used by readers in some way? Is the interested reader invited to build up his own data compilation by using the provided sources? Methods: The authors are very precise with their raw data sources (which is best practice and would allow interested readers to repeat their compiling of data), but they don't give as much detail on the methods they used to process the data and create the respective data layers. A brief description of the provided data layers is missing in the paper, e.g. in 2.4.1 (Zooplankton) it is not mentioned that via the persistent identifiers the maps with interpolated abundances of the two krill species can be accessed, but also a map with habitat suitability. Another example is chapter 2.4.2 (Zoobenthos), here species level data sources for asteroids, ophiuroids, and holothurians are listed, but the provided map layer shows one polygon only (special echinoderm assemblage). It should be made clear in each section what data products the user can access via the links, and how they are created. In my opinion the paragraphs on data sources could be shortened, as every source is listed also in table 1 and 2. Instead I would prefer to read more about the methods (e.g. models used).

Technical corrections Abstract line 29: Here it is spoken of five persistent identifiers, provided are six (also in 2.2). Suggest to sort the links according to the structure in the

paper (from abiotic to seals). p 5 line 11 f > a) b) c) in italics p 5 line 32 Possible to provide more detailed contact than institute webpage? References in text but missing in Reference section: Barthel & Gutt 1992, Timmermann 2013, Seitner et al. 2014 a, b, c (2004?)

---

## Referee Comment (RC2) · Anonymous Referee #2 · 7 Nov 2019

General comments

This paper provides a comprehensive data compilation of abiotic and biotic features of the Weddell Sea, relevant for the understanding of diverse ecosystem components that assist in the ongoing process for the development of an MPA in the region (hereafter WSMPA).

The paper is very well organized into major biodiversity groups of importance for the Weddell Sea region. It is also well written, clear, concise and short allowing the reader to focus on the most relevant aspects. The compilation of these data into a single manuscript is original and very useful for the process of developing a WSMPA and

for uses beyond it, as mentioned by the authors. Data is easily accessible in various formats, including high technical extensions (e.g. shapefiles, rasters; I used QGIS 3.2.3) and simple images, which make maps available for a larger audience. However, in my view, some information is missing, in particular related to metadata and methods description. These modifications could improve data accessibility, consistency with the text, geospatial information/outcomes accuracy and transparency of the wider WSMPA process.

Specific comments

1) Include further description of Methods used to analyze each data set and to develop each map. This could potentially be done in the paper itself as an Annex or in the Supplement section (including the maps), within the metadata file (adding an easier cross-link to the paper), and/or as a footnote/bigger caption in each available map. CCAMLR Working Groups or Workshops papers such as those submitted to EMM/SAM/WS are not generally available for the general public (login is required) so further information included therein should be available elsewhere for the interested reader. 2) In the description of the Methods, it could be good to include how the methods in each case were chosen (e.g. agreed by international community, based on specific paper, etc.) so it adds to the openness and transparency of the process. 3) It is not clear why only maps for 2.3 Environmental data are included at the end of the paper. I would suggest including maps (and methods) for 2.4 Biological data as well, for an easier and more comprehensive visualization 4) Avoid duplication of information in the text about data sources, references and cruise reports already included in the tables. 5) Most readers would probably be unfamiliar with CCAMLR. I would suggest adding a few general maps, including the CCAMLR Convention Area and the division in MPA Planning Domains (mentioned in the text) for contextualization. 6) In the 3. Outlook section, there is some mentioning to the development of a storage management system for this data. I would suggest also mentioning the CCAMLR MPA Information Repository (CMIR) that is under development by the CCAMLR Secretariat, as an additional suitable storage

space.

Technical corrections

i. Include CRS and projections information in each metadata file (common and thematic layers) for each shapefile and raster. ii. Provide clear cross-reference links between metadata description and available maps (names do not always coincide and it is hard to keep track to which description fits which map) iii. If possible, allow for the zip data to keep a clear file name referenced to the data they contain for easier identification when downloaded in folders (in particular for the "Data shapefile raster"). iv. Map legends in Figure 3 are very hard to read – make sure high definition maps are provided in final draft or make maps bigger. v. In section 2.2 Data availability, paragraph 10, there is the mention to five persistent identifiers. However, six of those are provided. Be aware that the same happens in the Abstract.

---

## Author Comment (AC1) · 25 Nov 2019

Anonymous Referee #1 - https://doi.org/105194/essd-2019-86-RC1

Specific comments

1. "At first reading of title and abstract I was under the impression that a huge data compilation is provided with this paper, while in fact a systematic overview of all the sources to the MPA planning process is given. This should be made more clear right from the begin, so the reader knows what to expect."

a. We agree with the referee's statement that we have to clarify right from the beginning

of the paper that a systematic overview of all the data sources - instead of a huge data compilation - used in the MPA planning process is given. We will change the whole text (incl. title) accordingly.

2. "Usability of data: The authors should elaborate if and how the provided data and information might be used, and/or to which future work they might contribute. Can the provided data layer products directly be used by readers in some way? Is the interested reader invited to build up his own data compilation by using the provided sources?" From general comments: "How or if the provided data layer products might be used, without access to the underlying data compilation, is not discussed."

a. We have mentioned in section "2.2 Data availability" (lines 15-16) that our data layer products can be used for geo-statistical analyses within the framework of MPA planning, among other things. Nevertheless, we will add some more details on data usability in the text in the appropriate sections (e.g., "2.2 Data availability", "3 Outlook").

3. "Methods: . . ., but they don't give as much detail on the methods they used to process the data and create the respective data layers. A brief description of the provided data layers is missing in the paper, e.g. in 2.4.1 (Zooplankton) it is not mentioned that via the persistent identifiers the maps with interpolated abundances of the two krill species can be accessed, but also a map with habitat suitability. Another example is chapter 2.4.2 (Zoobenthos), here species level data sources for asteroids, ophiuroids, and holothurians are listed, but the provided map layer shows one polygon only (special echinoderm assemblage). It should be made clear in each section what data products the user can access via the links, and how they are created."

a. We will follow the referee's suggestion by indicating in each subsection under "2.3 Environmental data" and "2.4 Ecological data" which data layer products can be accessed via the PANGAEA links and by adding as supplementary material the method by which the data were processed and the respective data layer was developed.

4. "In my opinion the paragraphs on data sources could be shortened, as every source
is listed also in table 1 and 2. Instead I would prefer to read more about the methods (e.g. models used)."

a. We agree with this statement, that we could shorten the sections "2.3 Environmental data" and "2.4 Ecological data" by avoiding duplication of information in the text about e.g., references to publications and cruise reports, explicitly listed in the tables, too. We will change the text accordingly.

b. We have already commented on this remark earlier in this reply (see #3a.). We will provide the information about the analytical methods (e.g. models) in the revised version.

Technical corrections

5. "Here it is spoken of five persistent identifiers, provided are six (also in 2.2). Suggest to sort the links according to the structure in the paper (from abiotic to seals)."

a. We change the text accordingly.

b. We follow the referee's suggestion by sorting the PANGAEA links according to the structure of section "2.3 Environmental data" and "2.4 Ecological data".

6. "p 5 line 11 f > a) b) c) in italics"

a. We change the writing style to "normal".

7. "p 5 line 32 Possible to provide more detailed contact than institute webpage?"

a. We will add more information about the data warehouse of the Thuenen Institute of Sea Fisheries where data on krill are stored.

8. "References in text but missing in Reference section: Barthel & Gutt 1992, Timmermann 2013, Seitner et al. 2014 a, b, c (2004?)

a. We include Barthel and Gutt (1992) in the "Reference" section because they are missing there. However, we do not include Timmermann (2013) in the "Reference"

section, because the correct reference is Haid and Timmermann (2013), referred to in section "2.3.3 FESOM data" and listed in "References". We change "Seiter et al. 2014a, b, c" in the text and Table 1 to "Seiter et al. 2004a, b, c".

---

## Author Comment (AC2) · 25 Nov 2019

Anonymous Referee #2 - https://doi.org/10.5194/essd-2019-86-RC2

Specific comments 1. "Include further description of Methods used to analyze each data set and to develop each map. This could potentially be done in the paper itself as an Annex or in the Supplement section (including the maps), within the metadata file (adding an easier crosslink to the paper), and/or as a footnote/bigger caption in each available map. CCAMLR Working Groups or Workshops papers such as those submitted to EMM/SAM/WS are not generally available for the general public (login is required) so further information included therein should be available elsewhere for the

interested reader." From general comments: "However, in my view, some information is missing, in particular related to metadata and methods description."

a. We agree with the referee's statement that the methods used to analyse each data set and to develop each map should be available for the interested reader. Therefore, we will describe the methods in the "Supplement" section (including the maps).

2. "In the description of the Methods, it could be good to include how the methods in each case were chosen (e.g. agreed by international community, based on specific paper, etc.) so it adds to the openness and transparency of the process."

a. We agree that a sentence about how the methods were chosen in each case increases the openness and transparency of the MPA planning process. We will add this information to the methods in the "Supplement" section accordingly.

3. "It is not clear why only maps for 2.3 Environmental data are included at the end of the paper. I would suggest including maps (and methods) for 2.4 Biological data as well, for an easier and more comprehensive visualization."

a. We showed all described data for "2.4 Ecological data" (point data) in Figure 1 and mapped in Figure 2 the raster data for "2.3.1 IBCSO data", "2.3.2 AMSR-E sea ice maps" and "2.3.3 FESOM data". We have refrained from presenting the environmental variables, which have only been used as explanatory variables in species distribution models (despite the description of these variables in the text).

b. Furthermore, we will provide each data layer product as a map in the "Supplement" section in the revised version (see also remark #1a earlier in this reply).

4. "Avoid duplication of information in the text about data sources, references and cruise reports already included in the tables."

a. We will change text sections "2.3 Environmental data" and "2.4 Ecological data" by deleting e.g., references to publications and cruise reports, explicitly listed in the tables, too.

5. "Most readers would probably be unfamiliar with CCAMLR. I would suggest adding a few general maps, including the CCAMLR Convention Area and the division in MPA Planning Domains (mentioned in the text) for contextualization."

a. We will follow the referee's suggestion by adding a map including the CCAMLR Convention Area and the MPA Planning Domains.

6. "In the 3. Outlook section, there is some mentioning to the development of a storage management system for this data. I would suggest also mentioning the CCAMLR MPA Information Repository (CMIR) that is under development by the CCAMLR Secretariat, as an additional suitable storage space."

a. We will add the information in "3 Outlook" that the CCAMLR MPA Information Repository (CMIR), currently being developed by the CCAMLR Secretariat, will also be available in the future as a suitable storage location for metadata.

Technical corrections

i. "Include CRS and projections information in each metadata file (common and thematic layers) for each shapefile and raster."

a. We will add CRS and projections information in the revised version under "2.2 Data availability".

b. Furthermore, CRS and projections information is supplied for each shape and raster file in "Source" under "Layer Properties" if you upload the file in GIS-software or open the ArcMAP packages.

c. In addition, projections information is named for each map in the legend (see folder "map_png").

ii. "Provide clear cross-reference links between metadata description and available maps (names do not always coincide and it is hard to keep track to which description fits which map)."

a. We follow the referee's suggestion by indicating for each metadata description (under "2.3 Environmental data" and "2.4 Ecological data"), which map was developed and under which file name the respective data layer is stored in PANGAEA.

iii. "If possible, allow for the zip data to keep a clear file name referenced to the data they contain for easier identification when downloaded in folders (in particular for the "Data shapefile raster")."

a. We have had the file names for zip data folders changed accordingly by the great support of the PANGAEA team. Because if a PANGAEA data entry is registered by a DOI, then - strictly speaking - nothing can be changed in the data publication anymore.

iv. "Map legends in Figure 3 are very hard to read – make sure high definition maps are provided in final draft or make maps bigger."

a. We agree that the map legends in Figure 3 are impossible to read. We change the maps accordingly.

v. "In section 2.2 Data availability, paragraph 10, there is the mention to five persistent identifiers. However, six of those are provided. Be aware that the same happens in the Abstract."

a. We change the text accordingly.

---

## Author Response (AR1)

**An integrated data compilationof data sources for the development of a marine protected area in the Weddell Sea**

Katharina Teschke1,2, Hendrik Pehlke1,2, Volker Siegel3, Horst Bornemann1, Rainer Knust1, Thomas Brey1,2,4

1 Alfred-Wegener-Institut, Helmholtz-Zentrum für Polar und Meeresforschung, Am Handelshafen 12, 27570 Bremerhaven, Germany

[revised manuscript text omitted]

The data described under 2.3.4 to 2.3.6 were used as explanatory variables in the Antarctic krill species distribution model (SDM) (SeaWiFS, WOA13, chemical sediment components) and in the demersal fish SDM (WOA13, chemical sediment components). The SDMs are described in detail in the Supplement and the PANGAEA link to the respective data layer products (incl. file names) is given in the corresponding subsection under "2.4 Ecological data".

**25 2.4 Ecological data**

In the following, we describe the sources of raw data sets used in the WSMPA planning process and indicate which data layer product was developed on the basis of which raw data sets per higher taxonomic group. In addition, the methods for processing and analysing the data and for developing each data layer are described in detail in the Supplement.

**30 2.4.1 Zooplankton**

**Antarctic krill (adults)**

The WSMPA data collection on adult Antarctic krill (*Euphausia superba*) originates from (i) historical UK data from "Discovery Expeditions" (1928-1939) and data collected during the SIBEX cruise by British Antarctic Survey, (ii) five South African data sets from the 1990s, (iii) four Soviet data sets from 1998 and 1990, (iv) Polish data (Witek et al., 1985) and (v) German data from location discovery cruises with MV "Polarsirkel" in

- 5 1979/80 and 1980/81 (Siegel, 1982), RV "Walther Herwig" cruises (1975/76, 1977/78) and the 2004 Lazarev Sea Krill Survey (LAKRIS) (RV "Polarstern" cruise ANT-XXI/4) (Siegel, 2012). All the data are publicly available via the database KRILLBASE (doi.org/brg8) (Atkinson et al., 2017) (see Table S2 in the Supplement that provides a detailed list of data used from KRILLBASE). The data from KRILLBASE were complemented by abundance data on *E. superba*, which were collected (a) during the Norwegian Antarctic research expedition
- 10 1976/77 (MV "Polarsirkel")-(Fevolden, 1979), (b) during two Soviet research cruises in 1977 (RV "Gizhiga") and 1983-(RV "Volny Vetter"), (c) in the context of the Lazarev Sea Krill Survey (RV "Polarstern" cruises ANT-XXIII/2, ANT-XXIII/6, ANT-XXIV/2) (e.g. Siegel, 2012) as well as (d) during RV "Polarstern" cruises ANT-V/1-3, ANT-VII/4, ANT-XVIII/4 and ANT-XXIX/3 (Siegel et al., 2013) (Table 2). Furthermore, Japanese, Norwegian and Soviet fisheries Fisheries data (eatch and effort) on *E. superba* for the WSMPA Planning Area
   15 (i.e. Statistical Subarea 48.5 and southern part of Subarea 48.6) stem from the CCAMLR database (data request

through CCAMLR Secretariat: 03-10-2013) (Table 2).

All these data were used in a species distribution model (SDM) of adult Antarctic krill and ultimately led to a data layer product showing habitat suitability for adult Antarctic krill in the WSMPA Planning Area (see doi 10.1594/PANGAEA.899667; file name: "Adult Antarctic krill, *Euphausia superba* - habitat suitability prediction").

**Antarctic krill (larvae)**

20

Abundance data on Antarctic krill larvae stem from (a) the Antarctic research expeditions 1976/77 (Fevolden, 1979)- and 1979/80 with MV "Polarsirkel"-(Siegel, 1982), (b) the First International BIOMASS Experiment
survey (FIBEX), (c) (RV "Walther Herwig" cruise 1981) (e.g. Trathan and Everson, 1994) and the Lazarev Sea Krill Survey (LAKRIS) (RV "Polarstern" cruises ANT-XXI/4, ANT-XXIII/6) (Siegel, 2012) as well as (ed) RV "Polarstern" cruise ANT-VII/4 and the combined RV "Polarstern" (ANT-VIII/2) and RV "Akademik Fedorov" cruises (Menshenina, 1992) (see Table 2).

All data on Antarctic krill larvae were used for an interpolation approach and led to a map of the interpolated 30 abundances of krill larvae in the WSMPA Planning Area (see doi 10.1594/PANGAEA.899667; "Antarctic krill larvae, *Euphausia superba* - interpolated abundance").

**Ice krill**

Abundance data on adult ice krill (*Euphausia crystallorophias*) originate from pelagic trawl surveys during (a)
 the German Antarctic research cruise 1975/76 with "Walther Herwig", (b) the "Pre-Site Survey" 1979/80 with MV "Polarsirkel" (Siegel, 1982), (c) the Lazarev Sea Krill Survey (RV "Polarstern" cruises ANT-XXI/4, ANT-XXIII/2, ANT-XXIII/6, ANT-XXIV/2) (e.g. Siegel, 2012)-as well as (d) RV "Polarstern" cruises ANT-V/1-3, ANT-VII/4 and ANT-XXIX/3 (Siegel et al., 2013) (Table 2).

The abundance data on *E. crystallorophias* were used for an interpolation approach and led to a map showing the
 interpolated abundances of ice krill (see doi 10.1594/PANGAEA.899667; "Ice krill, *Euphausia crystallorophias* - interpolated denisty"). In addition, the abundance data on *E. crystallorophias* were used for "ground truthing"

of the potential ice krill habitat (doi 10.1594/PANGAEA.899667; "Ice krill, *Euphausia crystallorophias* – pot habitat").

All data about *E. superba* and *E. crystallorophias*, which were used additionally to KRILLBASE and the CCAMLR database, are stored in the data warehouse of the Thuenen Institute of Sea Fisheries (https://www.thuenen.de; contact: Lara Kim Hühnerlage; kim.huenerlage@thuenen.de) and can be requested on demand.

**2.4.2 Zoobenthos**

**Sponges**

5

- 10 Abundance data and semi-quantitative data on sponges (higher taxonomic groups), which were compiled in the context of the WSMPA planning initiative, originate from zoobenthos data sets. The abundance data are publically available via PANGAEA (see Gerdes, 2014 a-o). The semi-quantitative data set can be requested from us if required (contact: Katharina Teschke, AWI) and is available as presence-absence data set in PANGAEA (see Teschke and Brey, 2019a) (see Table 2).
- 15 Based on these data, we developed a map of the occurrence of sponges in the WSMPA Planning Area (doi 10.1594/PANGAEA.899645; "Sponges, Porifera - interpolated presence").

**Echinoderms**

The data set on echinoderms consists of presence-absence data on species level for asteroids, abundance data on 20 ophiuroid taxa as well as holothurian taxa. The first two data sets are available in PANGAEA (Teschke and Brey, 2019b, c), the latter in the information system biodiversity.aq (Gutt et al., 2014). Publications, which have used those primary data sets, are e.g. Dahm (1996), Gutt (1988) and Gerdes et al. (1992).

These data were used in a clustering approach to ultimately identify the potential habitat for echinoderms in the WSMPA Planning Area by environmental proxies (doi 10.1594/PANGAEA.899645; "Special echinoderm assemblage - pot habitat").

**2.4.3 Fish**

25

**Antarctic silverfish (larvae and adults)**

The WSMPA data collection on Antarctic silverfish larvae (*Pleuragramma antarctica*) originates from quantitative zooplankton data sets obtained during the RV "Polarstern" cruises ANT-I/2 (Boysen-Ennen and Piatkowski, 1988) and ANT-III/3 (Hubold et al., 1988) and during the Lazarev Sea Krill Survey (LAKRIS) ("Polarstern" cruises: ANT XXI/4, ANT XXIII/6, ANT XXIV/2) (Flores et al., 2014) (Table 2). The first mentioned data are stored in the data warehouse of the Thuenen Institute of Sea Fisheries and can be requested on demand (contact: Lara Kim Hühnerlage). Fish larvae data from ANT-III/3 are available from Hubold et al. (1988) and the LAKRIS data can be requested from Hauke Flores (AWI) if required. are available in PANC AFA references will be added device proving proving.

35 PANGAEA (PANGAEA reference will be added during review process).

All abundance data on Antarctic silverfish (adults and larvae) were used for an interpolation approach and led to a map of the interpolated abundances of *P. antarctica* in the WSMPA Planning Area (doi 10.1594/PANGAEA.899591; "Antarctic silverfish, *Pleuragramma antarctica* - interpolated abundance").

**5 Demersal fish**

15

20

35

Abundance data on demersal fish and adult *P. antarctica* stem from benthic and pelagic trawl surveys during seven "Polarstern" cruises between 1996 and 2011 (ANT XIII/3, ANT XV/3, ANT XVII/3, ANT XIX/5, ANT-XXI/2, ANT XXII/8, ANT XXVII/3) (Table 2). and .- Publications, which have used these data, are e.g. Caecavo et al. (2018) and Mintenbeck et al. (2012). The primary data sets can be requested from us if required

10 (contact: Rainer Knust, AWI). –This data compilation was complemented by data on demersal fish and *P. antarctica* derived from trawl and dredge surveys published in PANGAEA (Drescher et., 2012; Ekau et al., 2012a, b; Hureau et al., 2012; Kock et al., 2012; Wöhrmann et al., 2012).

All data on demersal fish were used in a SDM and led to a data layer product showing the habitat suitability for demersal fish in the WSMPA Planning Area (see doi 10.1594/PANGAEA.899591; "Demersal fish - habitat suitability prediction").

Antarctic toothfish (adults)

Fishery data (catch per unit effort) on the Antarctic toothfish (*Dissostichus mawsoni*) for the WSMPA Planning Area (i.e. Statistical Subarea 48.5 and southern part of Subarea 48.6) were taken from the CCAMLR database and requested through the CCAMLR Secretariat (data request: 03-08-2016) (Table 2).

The data were used to determine the potential habitat of *D. mawsoni* in the WSMPA Planning Area (see doi 10.1594/PANGAEA.899591; "Adult toothfish, *Dissostichus mawsoni* - pot habitat").

**Demersal fish nesting sites**

Information about nesting sites of demersal fish was collected during the RV "Polarstern" cruises PS82 (Knust and Schröder, 2014) (ANT XXIX/9) and PS96 (Piepenburg, 2016) (ANT XXXI/2). The data are available from Knust and Schröder (2014) (PS82) and Piepenburg (2016) (PS96). The data collected during RV "Polarstern" cruises were supplemented by data from the literature (Daniels 1978, 1979; Jones & Near 2012). The map with the locations of the nesting sites of demersal fish is available at PANGAEA (doi 10.1594/PANGAEA.899591;
 "Demersal fish - observation of nesting sites") and is also shown in the Supplement (see Fig. S12).

**2.4.4 Flying and non-flying seabirds**

**Breeding and non-breeding Adélie penguins**

Tracking data on breeding\_and non-breeding Adélie penguins (*Pygoscelis adeliae*) originate from (i) British Antarctic Survey (BAS) inventory data from Phil Trathan (ID 754) and Mike Dunn and P. Trathan (ID 764 (only data on breeding Adélies), ID 773, 779), (ii) a data set from BAS (P. Trathan) and Instituto Antártico Argentino (Mercedes Santos) (ID 753) (Warwick et al., 2019) and (iii) a data set from the US AMLR Program from Jefferson Hinke and Wayne Trivelpiece (NOAA) (ID 910) (see e.g. Hinke et al. 2015) (see also Table 2). All the data are stored in the Birdlife International's Seabird Tracking Database (data request: 20-10-2015). Adélie penguins breeding locations and estimated abundances of breeding pairs were derived from Lynch and LaRue (2014).

The tracking data on *P. adeliae* were used to model the probability of breeding and non-breeding *P. adeliae* occurrence during foraging (doi 10.1594/PANGAEA.899520; "Breeding Adélie penguin, *Pygoscelis adeliae* -

5 modelled foraging trips" and "Non-breeding Adélie penguin, *Pygoscelis adeliae* - modelled foraging trips"). The final data layer product for breeding *P. adeliae* also depict breeding locations and estimated abundances of breeding pairs as well as buffer areas around each colony.

Tracking data on non-breeding *P. adeliae* were acquired from Birdlife International's Scabird Tracking
 Database, too (data request: 20-10-2015) (Table 2). Downloaded data include (i) BAS inventory data from Phil
 Trathan (ID 754) and Mike Dunn and P. Trathan (ID 773, 779), (ii) a data set from BAS (P. Trathan) and
 Instituto Antártico Argentino (Mercedes Santos) (ID 753) and (iii) a data set from the US AMLR Program from
 Jefferson Hinke and Wayne Trivelpiece (NOAA) (ID 910).

**15 Breeding Emperor penguins**

Data on Emperor penguin (*Aptenodytes forsteri*) colony locations and breeding population estimates were derived from Fretwell et al. (2012, 2014) (Table 2).

These data were used to develop a probability map of foraging areas for *A. forsteri* (doi 10.1594/PANGAEA.899520; "Breeding emperor penguin, *Aptenodytes forsteri* - modelled foraging areas").

**Antarctic petrels**

20

Information on breeding locations and estimated number of breeding pairs of the Antarctic petrel (*Thalassoica antarctica*) were kindly provided by Jan van Franeker (Wageningen University & Research) and are published in van Franeker et al. (1999) (Table 2).

25 The information on breeding pairs and their colony locations is shown in the final data layer product next to modelled foraging habitats of *T. antarctica* (doi 10.1594/PANGAEA.899520; "Antarctic petrel, *Thalassoica antarctica* - modelled foraging areas").

**2.4.5 Pinnipeds**

Tracking data from pinnipeds were obtained from the MEOP data portal "Marine Mammals Exploring the Oceans Pole to Pole" (data request: 14-11-2016) (see Table 2 for a detailed list of data used). In addition, we have used MEOP data (UK data: ct27, ct70; German data: ct113, wd06, wd07) for which unconditional sharing were not yet accepted at the time of data retrieval and were provided by Lars Boehme (University of St. Andrews) and us (H. Bornemann), respectively. The UK and German data sets are now also freely accessible from the MEOP data portal. Furthermore, the data from the MEOP data portal were complemented by tracking data sets on southern elephant seals (Tosh et al., 2009a, b: James et al., 2012a, b). Weddell seals (McIntvre et al., 2015a)

trom the MEOP data portal. Furthermore, the data from the MEOP data portal were complemented by tracking data sets on southern elephant seals (Tosh et al., 2009a, b; James et al., 2012a, b), Weddell seals (McIntyre et al., 2013a, b) and crabeater seals (Nachtsheim et al., 2016a, b) stored in PANGAEA. All these tracking data were used to model the probability of seal occurrence during foraging (doi

All these tracking data were used to model the probability of seal occurrence during foraging (do 10.1594/PANGAEA.899619; "Seal abundance - modelled prediction values").

Point data from pack-ice seals (unspecified taxa) based on aerial surveys are from Plötz et al. (2011a-e) and were downloaded from PANGAEA (Table 2). These data were sampled during five flight campaigns from 1996 to 2001 within the Antarctic Pack Ice Seals (APIS) programme. In addition, information on crabeater seal densities (predicted or observed) was derived from Bester et al. (1995 and 2002), Flores et al. (2008) and Forcada et al.

(2012; Table 2). German and South African APIS data and UK census data were published in e.g. Gurarie et al. (2016) and Foreada et al. (2012), respectively.

All the APIS point data and information on seal densities were used to develop a map showing the distribution patterns of seals in the WSMPA Planning Area (10.1594/PANGAEA.899619; "Seal abundance - modelled and interpolated prediction values").

**10 3 Outlook**

5

This is the first data-compilationof data sources for the Antarctic Weddell Sea and adjacent seas, which considers data across the entire ecosystem: i.e., from abiotic data, such as bathymetry and sea ice, to ecological data ranging from zooplankton and zoobenthos to fish, birds and marine mammals. The effort to create such a data-compilationof data sources was directly coupled with the initiative to develop a WSMPA. However, our

- 15 compilation of data sources will facilitate the future research on fauna, ecology and nature conservation in the Weddell Sea. Using our systematic overview of available data for the development of a specific data collection, future projects save the time-consuming multi-parameter data search from the scratch. In addition, our work serves to guide future studies aimed at closing data gaps in the wider Weddell Sea region and/or simply pointing to specific data sets that may be of particular interest to future generations (baseline is a particular issue). For
- 20 example, However, the data compilation is also suitable for further scientific questions in the wide field of faunistic, ecological and nature conservation studies to investigate the effect of climate change and possible fishing activities in this area. Ssome of the ecological data sets were collected in the 1980s and earlier, when the Weddell Sea was still almost pristine and hardly affected by any anthropogenic activities, so that these data sets are optimally suited to describe a reference state for assessing the effect of pressures on the Weddell Sea
- 25 ecosystem. In addition, the ecological data with a few exceptions provide information on abundances of the respective taxa and are therefore better suited as an indicator for environmental changes than presence-absence data or presence data only.

Ultimately, the data-compilation of data sources serves to protect our data heritage for use by future generations (baseline is a particular issue), to enable work with readily available multi-parameter data sets, and to-motivate

- 30 researches researchers to add-incorporate further data, both from existing "paper sources" and from future measurements, into existing data repositories and archives.
- Subsequent work will focus on the development of an efficient and tailor-made management system for the storage of these complex and heterogeneous data and information of WSMPA data compilation and automated data mining, handling and analysis. This system will serve three purposes: (i) to better enable a more holistic and integrative approach towards ecosystem research in the Weddell Sea in general, (ii) to enable the management of the WSMPA to carry out the tasks of the Research and Monitoring Programme as a mandatory part of an MPA under CCAMLR when adopting the MPA, and (iii) to provide key stakeholders and the public with access to data, information and management measures related to the ecosystem of the Weddell Sea region in general and the WSMPA in particular. The CCAMLR MPA Information Repository (CMIR) currently being developed by

the CCAMLR Secretariat will also be available in the future as a suitable storage location for metadata on CCAMLR MPAs in Antarctica.

Author contribution. KT collected all data together, described the metadata and led the writing of the paper. HP took over the technical part of the data acquisition (retrieval, storage, processing). VS collected and prepared the data on zooplankton for further analyses within the WSMPA planning. HB and RK were significantly involved in the collection of the data on pinnipeds and fishes, respectively. TB collaborated in the paper writing.

Competing interests. The authors declare that they have no conflict of interests.

Acknowledgements. This work was financially supported by the German Federal Ministry of Food and Agriculture (BMEL) through the Federal Office for Agriculture and Food (BLE), grant number 2813HS009. In
particular, we would like to thank all colleagues from all national and international scientific institutions who have supported us in providing data used to build up the data compilation for the wider Weddell Sea: i.e. from the Alfred Wegener Institute (Hauke Flores, Dieter Gerdes, Julian Gutt, Stefan Hain, Kerstin Jerosch, Rainer Knust, Dieter Piepenburg, Ralf Timmermann), British Antarctic Survey (Phil Trathan), CCAMLR Secretariat (Elanor Miller, Tim Jones, David Ramm), Helmholtz Centre Geesthacht (Verena Haid), Instituto Antártico

- 15 Argentino (Mercedes Santos), National Oceanic and Atmospheric Administration (Jefferson Hinke), Royal Belgian Institute of Natural Sciences (Anton van de Putte), Stony Brook University (Heather Lynch), Thünen Institute of Sea Fisheries (Karl-Hermann Kock), University of Gothenburg (Tomas Lundälv), University of Padova (Emilio Riginella), University of St. Andrews (Lars Boehme), Wageningen University & Research (Jan van Franeker). The marine mammal data were collected and made freely available by the International MEOP
- 20 Consortium and the national programs that contribute to it (http://www.meop.net). The seal tracking data ct96 and ct109 are collected by the Integrated Marine Observing System (IMOS). IMOS is a national collaborative research infrastructure, supported by the Australian Government. It is operated by a consortium of institutions as an unincorporated joint venture, with the University of Tasmania as Lead Agent. We thank two anonymous reviewers for careful reading and constructive comments on the manuscript.

25

5

Figure 1.

l

---

## Author Response (AR2)

**Reply to public comments from the topical editor**

Dear Mr Fleischer,

We thank you for your report on our updated version of the manuscript entitled *"An integrated compilation of data sources for the development of a marine protected area in the Weddell Sea"*.

We have revised our manuscript according to your recommendations, i.e. we have made the following changes to the manuscript:

1) We have changed the PANGAEA internal DOI resolver to the official DOI resolver, i.e. https://doi.org/, in Table 1 and Table 2 (see pp. 14-19).

2) All data sets from our group of authors are now published in the data repository PANGAEA, i.e. (a) the semi-quantitative dataset on zoobenthos (p. 6) and (b) the datasets on fishes (p. 7). Please note that the DOI resolver of the data collection "fishes" is already officially registered (https://doi.org/10.1594/PANGAEA.911972), but the individual data sets on fish are still "in review" and password protected until the PANGAEA team uploads the data. You can view the data with this temporary access key, which was created especially for us: https://www.pangaea.de/tok/f07d0042ef0360f809c916870b7371150d03c4ae.

3) We have written an additional paragraph in the Supplementary Material, which focuses on data preparation and cleaning.

Please see the changes - we did as track changes - in the main text and the Supplementary Material.

Kind regards,
Katharina Teschke and co-authors

**Reply to non-public comments from the topical editor**

Dear Mr Fleischer,

We thank you for your offer to participate in the ESSD pilot project to publish our R scripts. We are fully aware of the importance of reproducibility and thus of the greater transparency of scientific work. Especially in our study area, which is located at the interface between science and politics, transparency of work is a basic requirement for successful negotiations, e.g. on the establishment of MPAs. Therefore, efforts are already underway to publish our R scripts as packages in the GitHub or CRAN environment.

Best regards,
Katharina

**An integrated compilation of data sources for the development of a marine protected area in the Weddell Sea**

[revised manuscript text omitted]

20    The Weddell Sea is - despite being one of the most remote and inaccessible places on earth - relatively well investigated compared to other Antarctic regions.

Since approximately 30 years the Weddell Sea is the geographical focus area of the German Antarctic research. In addition, there are manifold research activities of other nations. Consequently, we were able to compile a tremendous amount of environmental and ecological data to support the development of a Weddell Sea MPA

25    (hereafter: WSMPA) under the Commission for the Conservation of Antarctic Marine Living Ressources (CCAMLR). Here we present a systematic overview of all environmental and ecological data sources collected for the development of a WSMPA and provide data layer products that are based on this data compilation.

**2  Data description**

**2.1 Study site**

[revised manuscript text omitted]

The data described under 2.3.4 to 2.3.6 were used as explanatory variables in the Antarctic krill species distribution model (SDM) (SeaWiFS, WOA13, chemical sediment components) and in the demersal fish SDM (WOA13, chemical sediment components). The SDMs are described in detail in the Supplement and the PANGAEA link to the respective data layer products (incl. file names) is given in the corresponding subsection under "2.4 Ecological data".

**2.4 Ecological data**

In the following, we describe the sources of raw data sets used in the WSMPA planning process and indicate which data layer product was developed on the basis of which raw data sets per higher taxonomic group. In addition, the methods for processing and analysing the data and for developing each data layer are described in detail in the Supplement.

**2.4.1 Zooplankton**

**Antarctic krill (adults)**

The WSMPA data collection on adult Antarctic krill (*Euphausia superba*) originates from (i) historical UK data from "Discovery Expeditions" (1928-1939) and data collected during the SIBEX cruise by British Antarctic Survey, (ii) five South African data sets from the 1990s, (iii) four Soviet data sets from 1998 and 1990, (iv) Polish data (Witek et al., 1985) and (v) German data from location discovery cruises with MV "Polarsirkel" in 1979/80 and 1980/81 (Siegel, 1982), RV "Walther Herwig" cruises (1975/76, 1977/78) and the 2004 Lazarev Sea Krill Survey (LAKRIS) (RV "Polarstern" cruise ANT-XXI/4) (Siegel, 2012). All the data are publicly available via the database KRILLBASE (doi.org/brg8) (Atkinson et al., 2017) (see Table S2 in the Supplement that provides a detailed list of data used from KRILLBASE). The data from KRILLLBASE were complemented by abundance data on *E. superba*, which were collected (a) during the Norwegian Antarctic research expedition 1976/77 (MV "Polarsirkel"), (b) during two Soviet research cruises in 1977 and 1983, (c) in the context of the Lazarev Sea Krill Survey  as well as (d) during RV "Polarstern" cruises ANT-V/1-3, ANT-VII/4, ANT-XVIII/4

and ANT-XXIX/3 (Table 2). Furthermore, Fisheries data on *E. superba* for the WSMPA Planning Area (i.e. Statistical Subarea 48.5 and southern part of Subarea 48.6) stem from the CCAMLR database (data request through CCAMLR Secretariat: 03-10-2013) (Table 2).

All these data were used in a species distribution model (SDM) of adult Antarctic krill and ultimately led to a data layer product showing habitat suitability for adult Antarctic krill in the WSMPA Planning Area (see doi 10.1594/PANGAEA.899667; file name: "Adult Antarctic krill, *Euphausia superba* - habitat suitability prediction").

**Antarctic krill (larvae)**

Abundance data on Antarctic krill larvae stem from (a) the Antarctic research expeditions 1976/77 and 1979/80 with MV "Polarsirkel", (b) the First International BIOMASS Experiment survey (FIBEX), (c) the Lazarev Sea Krill Survey (LAKRIS) as well as (d) RV "Polarstern" cruise ANT-VII/4 and the combined RV "Polarstern" (ANT-VIII/2) and RV "Akademik Fedorov" cruises (see Table 2).

All data on Antarctic krill larvae were used for an interpolation approach and led to a map of the interpolated abundances of krill larvae in the WSMPA Planning Area (see doi 10.1594/PANGAEA.899667; "Antarctic krill larvae, *Euphausia superba* - interpolated abundance").

**Ice krill**

Abundance data on adult ice krill (*Euphausia crystallorophias*) originate from pelagic trawl surveys during (a) the German Antarctic research cruise 1975/76 with "Walther Herwig", (b) the "Pre-Site Survey" 1979/80 with MV "Polarsirkel", (c) the Lazarev Sea Krill Survey as well as (d) RV "Polarstern" cruises ANT-V/1-3, ANT-VII/4 and ANT-XXIX/3 (Table 2).

The abundance data on *E. crystallorophias* were used for an interpolation approach and led to a map showing the interpolated abundances of ice krill (see doi 10.1594/PANGAEA.899667; "Ice krill, *Euphausia crystallorophias* - interpolated denisty"). In addition, the abundance data on *E. crystallorophias* were used for "ground truthing" of the potential ice krill habitat (doi 10.1594/PANGAEA.899667; "Ice krill, *Euphausia crystallorophias* – pot habitat").

All data about *E. superba* and *E. crystallorophias,* which were used additionally to KRILLBASE and the CCAMLR database, are stored in the data warehouse of the Thuenen Institute of Sea Fisheries (https://www.thuenen.de; contact: Lara Kim Hühnerlage; kim.huenerlage@thuenen.de) and can be requested on demand.

**2.4.2 Zoobenthos**

**Sponges**

Abundance data and semi-quantitative data on sponges (higher taxonomic groups), which were compiled in the context of the WSMPA planning initiative, originate from zoobenthos data sets. The abundance data (Gerdes, 2014 a-o) and the semi-quantitative data (Teschke and Brey, 2020) are publically available via PANGAEA (see Gerdes, 2014 a-o). The semi-quantitative data set can be requested from us if required (contact: Katharina

 (see Table 2).

Based on these data, we developed a map of the occurrence of sponges in the WSMPA Planning Area (doi 10.1594/PANGAEA.899645; "Sponges, Porifera - interpolated presence").

**Echinoderms**

The data set on echinoderms consists of presence-absence data on species level for asteroids, abundance data on ophiuroid taxa as well as holothurian taxa. The first two data sets are available in PANGAEA (Teschke and Brey, 2019a, b), the latter in the information system biodiversity.aq (Gutt et al., 2014).

10 These data were used in a clustering approach to ultimately identify the potential habitat for echinoderms in the WSMPA Planning Area by environmental proxies (doi 10.1594/PANGAEA.899645; "Special echinoderm assemblage - pot habitat").

**2.4.3 Fish**

**Antarctic silverfish and demersal fish**

15 The WSMPA data collection on Antarctic silverfish larvae (*Pleuragramma antarctica*) originates from quantitative zooplankton data sets obtained during the RV "Polarstern" cruises ANT-I/2 and ANT-III/3 and during the Lazarev Sea Krill Survey (LAKRIS) (Table 2). The first mentioned data (ANT-I/2) are stored in the data warehouse of the Thuenen Institute of Sea Fisheries and can be requested on demand (contact: Lara Kim Hühnerlage). Fish larvae data from ANT-III/3 are available from Hubold et al. (1988) and the LAKRIS data can

20 be requested from Hauke Flores (AWI) if required. Abundance data on demersal fish and adult *P. antarctica* stem from benthic and pelagic trawl surveys from six "Polarstern" cruises between 1996 and 2011 (Table 2), and  are published in PANGAEA  ( Knust 2020 or references therein, i.e. Balguerías and Knust, 2020; Knust et al., 2020a-d; Schröder and Knust, 2020). This data compilation was complemented by data on demersal fish and *P. antarctica* derived from trawl and dredge

25 surveys published in PANGAEA (Drescher et., 2012; Ekau et al., 2012a, b; Hureau et al., 2012; Kock et al., 2012; Wöhrmann et al., 2012).

All abundance data on Antarctic silverfish (adults and larvae) were used for an interpolation approach and led to a map of the interpolated abundances of *P. antarctica* in the WSMPA Planning Area (doi 10.1594/PANGAEA.899591; "Antarctic silverfish, *Pleuragramma antarctica* - interpolated abundance").

30 All data on demersal fish were used in a SDM and led to a data layer product showing the habitat suitability for demersal fish in the WSMPA Planning Area (see doi 10.1594/PANGAEA.899591; "Demersal fish - habitat suitability prediction").

**Antarctic toothfish (adults)**

35 Fishery data on the Antarctic toothfish (*Dissostichus mawsoni*) for the WSMPA Planning Area (i.e. Statistical Subarea 48.5 and southern part of Subarea 48.6) were taken from the CCAMLR database and requested through the CCAMLR Secretariat (data request: 03-08-2016) (Table 2).

The data were used to determine the potential habitat of *D. mawsoni* in the WSMPA Planning Area (see doi 10.1594/PANGAEA.899591; "Adult toothfish, *Dissostichus mawsoni* - pot habitat").

**Demersal fish nesting sites**

Information about nesting sites of demersal fish was collected during the RV "Polarstern" cruises PS82 (Knust and Schröder, 2014) and PS96 (Piepenburg, 2016). The data collected during RV "Polarstern" cruises were supplemented by data from the literature (Daniels 1978, 1979; Jones and Near 2012). The map with the locations of the nesting sites of demersal fish is available at PANGAEA (doi 10.1594/PANGAEA.899591; "Demersal fish - observation of nesting sites") and is also shown in the Supplement (see Fig. S12).

**2.4.4 Flying and non-flying seabirds**

**Breeding and non-breeding Adélie penguins**

Tracking data on breeding and non-breeding Adélie penguins *(Pygoscelis adeliae)* originate from (i) British Antarctic Survey (BAS) inventory data from Phil Trathan (ID 754) and Mike Dunn and P. Trathan (ID 764 (only data on breeding Adélies), ID 773, 779), (ii) a data set from BAS (P. Trathan) and Instituto Antártico Argentino (Mercedes Santos) (ID 753) and (iii) a data set from the US AMLR Program from Jefferson Hinke and Wayne Trivelpiece (NOAA) (ID 910) (see also Table 2). All the data are stored in the Birdlife International`s Seabird Tracking Database (data request: 20-10-2015). Adélie penguins breeding locations and estimated abundances of breeding pairs were derived from Lynch and LaRue (2014).

The tracking data on *P. adeliae* were used to model the probability of breeding and non-breeding *P. adeliae* occurrence during foraging (doi 10.1594/PANGAEA.899520; "Breeding Adélie penguin, *Pygoscelis adeliae* - modelled foraging trips" and "Non-breeding Adélie penguin, *Pygoscelis adeliae* - modelled foraging trips"). The final data layer product for breeding *P. adeliae* also depict breeding locations and estimated abundances of breeding pairs as well as buffer areas around each colony.

**Breeding Emperor penguins**

Data on Emperor penguin *(Aptenodytes forsteri)* colony locations and breeding population estimates were derived from Fretwell et al. (2012, 2014) (Table 2).

These data were used to develop a probability map of foraging areas for *A. forsteri* (doi 10.1594/PANGAEA.899520; "Breeding emperor penguin, *Aptenodytes forsteri* - modelled foraging areas").

**Antarctic petrels**

Information on breeding locations and estimated number of breeding pairs of the Antarctic petrel *(Thalassoica antarctica)* were kindly provided by Jan van Franeker (Wageningen University & Research) and are published in van Franeker et al. (1999) (Table 2).

The information on breeding pairs and their colony locations is shown in the final data layer product next to modelled foraging habitats of *T. antarctica* (doi 10.1594/PANGAEA.899520; "Antarctic petrel, *Thalassoica antarctica* - modelled foraging areas").

**2.4.5 Pinnipeds**

Tracking data from pinnipeds were obtained from the MEOP data portal "Marine Mammals Exploring the Oceans Pole to Pole" (data request: 14-11-2016) (see Table 2 for a detailed list of data used). In addition, we have used MEOP data (UK data: ct27, ct70; German data: ct113, wd06, wd07) for which unconditional sharing
5    were not yet accepted at the time of data retrieval and were provided by Lars Boehme (University of St. Andrews) and us (H. Bornemann), respectively. The UK and German data sets are now also freely accessible from the MEOP data portal. Furthermore, the data from the MEOP data portal were complemented by tracking data sets on southern elephant seals (Tosh et al., 2009a, b; James et al., 2012a, b), Weddell seals (McIntyre et al., 2013a, b) and crabeater seals (Nachtsheim et al., 2016a, b) stored in PANGAEA.
10   All these tracking data were used to model the probability of seal occurrence during foraging (doi 10.1594/PANGAEA.899619; "Seal abundance - modelled prediction values").

Point data from pack-ice seals (unspecified taxa) based on aerial surveys are from Plötz et al. (2011a-e) and were downloaded from PANGAEA (Table 2). These data were sampled during five flight campaigns from 1996 to
15   2001 within the Antarctic Pack Ice Seals (APIS) programme. In addition, information on crabeater seal densities (predicted or observed) was derived from Bester et al. (1995 and 2002), Flores et al. (2008) and Forcada et al. (2012; Table 2).
All the APIS point data and information on seal densities were used to develop a map showing the distribution patterns of seals in the WSMPA Planning Area (doi 10.1594/PANGAEA.899619; "Seal abundance - modelled
20   and interpolated prediction values").

**3 Outlook**

This is the first compilation of data sources for the Antarctic Weddell Sea and adjacent seas, which considers data across the entire ecosystem: i.e., from abiotic data, such as bathymetry and sea ice, to ecological data ranging from zooplankton and zoobenthos to fish, birds and marine mammals. The effort to create such a
25   compilation of data sources was directly coupled with the initiative to develop a WSMPA. However, our compilation of data sources will facilitate the future research on fauna, ecology and nature conservation in the Weddell Sea. Using our systematic overview of available data for the development of a specific data collection, future projects save the time-consuming multi-parameter data search from the scratch. In addition, our work serves to guide future studies aimed at closing data gaps in the wider Weddell Sea region and/or simply pointing
30   to specific data sets that may be of particular interest to future generations (baseline is a particular issue). For example, some of the ecological data sets were collected in the 1980s and earlier, when the Weddell Sea was still almost pristine and hardly affected by any anthropogenic activities, so that these data sets are optimally suited to describe a reference state for assessing the effect of pressures on the Weddell Sea ecosystem. In addition, the ecological data - with a few exceptions - provide information on abundances of the respective taxa and are
35   therefore better suited as an indicator for environmental changes than presence-absence data or presence data only.
Ultimately, the compilation of data sources serves to motivate researchers to incorporate further data, both from existing "paper sources" and from future measurements, into existing data repositories and archives.

Subsequent work will focus on the development of an efficient and tailor-made management system for the storage of these complex and heterogeneous data and information of WSMPA data compilation and automated data mining, handling and analysis. This system will serve three purposes: (i) to better enable a more holistic and integrative approach towards ecosystem research in the Weddell Sea in general, (ii) to enable the management of the WSMPA to carry out the tasks of the Research and Monitoring Programme as a mandatory part of an MPA under CCAMLR when adopting the MPA, and (iii) to provide key stakeholders and the public with access to data, information and management measures related to the ecosystem of the Weddell Sea region in general and the WSMPA in particular. The CCAMLR MPA Information Repository (CMIR) currently being developed by the CCAMLR Secretariat will also be available in the future as a suitable storage location for metadata on CCAMLR MPAs in Antarctica.

**Author contribution.** KT collected all data together, described the metadata and led the writing of the paper. HP took over the technical part of the data acquisition (retrieval, storage, processing). VS collected and prepared the data on zooplankton for further analyses within the WSMPA planning. HB and RK were significantly involved in the collection of the data on pinnipeds and fishes, respectively. TB collaborated in the paper writing.

**Competing interests.** The authors declare that they have no conflict of interests.

Acknowledgements. This work was financially supported by the German Federal Ministry of Food and Agriculture (BMEL) through the Federal Office for Agriculture and Food (BLE), grant number 2813HS009. In particular, we would like to thank all colleagues from all national and international scientific institutions who have supported us in providing data used to build up the data compilation for the wider Weddell Sea: i.e. from the Alfred Wegener Institute (Hauke Flores, Dieter Gerdes, Julian Gutt, Stefan Hain, Kerstin Jerosch, Rainer Knust, Dieter Piepenburg, Ralf Timmermann), British Antarctic Survey (Phil Trathan), CCAMLR Secretariat (Elanor Miller, Tim Jones, David Ramm), Helmholtz Centre Geesthacht (Verena Haid), Instituto Antártico Argentino (Mercedes Santos), National Oceanic and Atmospheric Administration (Jefferson Hinke), Royal Belgian Institute of Natural Sciences (Anton van de Putte), Stony Brook University (Heather Lynch), Thünen Institute of Sea Fisheries (Karl-Hermann Kock), University of Gothenburg (Tomas Lundälv), University of Padova (Emilio Riginella), University of St. Andrews (Lars Boehme), Wageningen University & Research (Jan van Franeker). The marine mammal data were collected and made freely available by the International MEOP Consortium and the national programs that contribute to it (http://www.meop.net). The seal tracking data ct96 and ct109 are collected by the Integrated Marine Observing System (IMOS). IMOS is a national collaborative research infrastructure, supported by the Australian Government. It is operated by a consortium of institutions as an unincorporated joint venture, with the University of Tasmania as Lead Agent. We thank two anonymous reviewers for careful reading and constructive comments on the manuscript.

**Figure 1.**

[Figure]

**Figure 1.** CCAMLR Convention Area with its Marine Protected Area (MPA) Planning Domains and the planning area (incl. 200 km wide buffer area near the Antarctic Peninsula) for the development of a MPA in the wider Weddell Sea (red shaded area). Domain 1: Western Peninsula - South Scotia Arc, Domain 2: North Scotia Arc, Domain 3: Weddell Sea, Domain 4: Bouvet Maud, Domain 5: Crozet - del Cano, Domain 6: Kerguelen Plateau, Domain 7: Eastern Antarctica, Domain 8: Ross Sea, Domain 9: Amundsen - Bellingshausen.

**Figure 2.**

[revised manuscript text omitted]

| | | | | | |
|---|---|---|---|---|---|
| Sponges (presence-absence) | ANT-VII/4 ANT-IX/3 ANT-XIII/3 ANT-XV/3 ANT-XXI/2 | Teschke and Brey (2020) | Arntz et al. (1990) Bathmann et al. (1992) Arntz and Gutt (1997) Arntz and Gutt (1999) Arntz and Brey (2005) | Galéron et al. (1992) | https://doi.org/10.1594/PANGAEA.899645 |
| Echinoderms - Asteroids (presence-absence) | ANT-I/2 ANT-II/4 ANT-V/3 ANT-VI/3 ANT-XV/3 ANT-XVII/3 | Teschke and Brey (2019a) | Drescher et al. (1983) Kohnen (1984) Schnack-Schiel (1987) Fütterer (1988) Arntz and Gutt (1999) Arntz and Brey (2001) | Voß (1988) | https://doi.org/10.1594/PANGAEA.899645 |
| Echinoderms - Ophiuroids (abundances) | ANT-I/2 ANT-II/4 ANT-V/3 ANT-V/4 ANT-VI/3 ANT-VII/4 ANT-IX/3 ANT-X/3 | Teschke and Brey (2019b) | Drescher et al. (1983) Kohnen (1984) Schnack-Schiel (1987) Miller and Oerter (1990) Fütterer (1988) Arntz et al. (1990) Bathmann et al. (1992) Spindler et al. (1993) | Brey et al. (1994) Dahm (1996) | https://doi.org/10.1594/PANGAEA.899645 |
| Echinoderms - Holothurians (abundances) | ANT-I/2 ANT-II/4 ANT-III/3 | Gutt, Piepenburg and Voß (2014) | Drescher et al. (1983) Kohnen (1984) Hempel (1985) | Gutt (1988) Piepenburg et al. (1997) | https://doi.org/10.1594/PANGAEA.899645 |

**Fishes**

| | | | | | |
|---|---|---|---|---|---|
| Fish larvae - *Pleuragramma antarctica* (abundances) | ANT-I/2 | Database of Thuenen Institute of Sea Fisheries | Drescher et al (1983) | Boysen-Ennen and Piatkowski (1988) | https://doi.org/10.1594/PANGAEA.899591 |
| Fish larvae - *Pleuragramma antarctica* (abundances) | ANT-III/3 | Hubold et al. (1988) | Hempel (1985) | | https://doi.org/10.1594/PANGAEA.899591 |
| Fish larvae - *Pleuragramma antarctica* (abundances) | Lazarev Sea Krill Survey (LAKRIS) data: ANT-XXI/4, ANT-XXIII/6, ANT-XXIV/2 | Contact: Hauke Flores (AWI) Hauke.Flores@awi.de | Smetacek et al. (2005) Bathmann (2008, 2010) | Flores et al. (2014) | https://doi.org/10.1594/PANGAEA.899591 |

| Data | Cruises/Source | References (red/underlined = new) | References | References | DOI |
|---|---|---|---|---|---|
| *Pleuragramma antarctica* & demersal fishes (abundances) | ANT-XIII/3 ANT-XV/3 ANT-XVII/3 ANT-XIX/5 ANT-XXI/2 ANT-XXVII/3 | Balguerías and Knust (2020) Knust and Schröder (2020) Knust et al. (2020a) Knust et al. (2020b) Knust et al. (2020c) Knust et al. (2020d) Contact: Rainer Knust (AWI) Rainer.Knust@awi.de | Arntz and Gutt (1997) Arntz and Gutt (1999) Arntz and Brey (2001) Arntz and Brey (2003) Arntz and Brey (2005) Knust et al. (2012) | Mintenbeck et al. (2012) Caccavo et al. (2018) | https://doi.org/10.1594/PANGAEA.899591 |
| *Pleuragramma antarctica* & demersal fishes (abundances) | ANT-I/2 ANT-III/3 ANT-V/3 ANT-VII/4 ANT-IX/3 ANT-XXIII/8 | Drescher et al. (2012) Ekau et al. (2012a) Ekau et al. (2012b) Hureau et al. (2012) Wöhrmann et al. (2012) Kock et al. (2012) | Drescher et al (1983) Hempel (1985) Schnack-Schiel (1987) Arntz et al. (1990) Bathmann et al. (1992) Gutt (2008) | Ekau (1988) Caccavo et al. (2018) | https://doi.org/10.1594/PANGAEA.899591 |
| Antarctic toothfish (catch per unit effort) | Japanese, Korean, Norwegian and South African fishing data | CCAMLR database; Contact: CCAMLR Secretariat [data request: 03-08-2016] | | | https://doi.org/10.1594/PANGAEA.899591 |
| Demersal fish nesting sites | PS82 (ANT-XXIX/9) PS96 (ANT-XXXI/2) | Knust and Schröder (2014) Piepenburg (2016) | Schröder (2016) | La Mesa et al. (2019) | https://doi.org/10.1594/PANGAEA.899591 |
| Demersal fish nesting sites | | Daniels (1978, 1979) Jones and Near (2012) | | | https://doi.org/10.1594/PANGAEA.899591 |

**Birds**

| Data | Cruises/Source | References | References | References | DOI |
|---|---|---|---|---|---|
| Adélie penguin colonies (estimated abundances of breeding pairs) | | Lynch and LaRue (2014) | | | https://doi.org/10.1594/PANGAEA.899520 |
| Breeding and non-breeding Adélie penguins (tracking data) | US AMLR Program (ID 910) | Birdlife International`s Seabird Tracking Database [data request: 20-10-2015] | | Hinke et al. (2015) | https://doi.org/10.1594/PANGAEA.899520 |
| Breeding and non-breeding Adélie penguins (tracking data) | BAS / Instituto Antártico Argentino data (ID 753) | Birdlife International`s Seabird Tracking Database [data request: 20-10-2015] | | Warwick-Evans et al. (2019) | https://doi.org/10.1594/PANGAEA.899520 |
| Breeding and non-breeding Adélie penguins (tracking data) | BAS Inventory (754, 773, 779) | Birdlife International`s Seabird Tracking Database [data request: 20-10-2015] | | Dunn et al. (2011) | https://doi.org/10.1594/PANGAEA.899520 |
| Breeding Adélie penguins (tracking data) | BAS Inventory (ID 764) | Birdlife International`s Seabird Tracking Database [data request: 20-10-2015] | | Lynnes et al. (2002) | https://doi.org/10.1594/PANGAEA.899520 |

| | | | | |
|---|---|---|---|---|
| Emperor penguin colonies (populations estimates) | | Fretwell et al. (2012)
Fretwell et al. (2014) | | https://doi.org/10.1594/PANGAEA.899520 |
| Antarctic petrel Colonies (estimated number of breeding pairs) | | Van Franeker et al. (1999) | | https://doi.org/10.1594/PANGAEA.899520 |
| **Pinnipeds** | | | | |
| Seal taxa (tracking data) | Data from: Australia (ct109, ct96), Brazil (ct56, ct46, ct39, ct22), China (ct105), UK (ct1, ct8, ct27, ct27x, ct40, ct43, ct45, ct49, ct58, ct70), France (ct16, ct62, ft01, ft02, ft11, ft12), Germany (ct21, ct35, ct35b, ct44, ct54, ct68, ct87, ct99, ct102, ct113, wd06, wd07), Norway (ct34), South Africa (ct33, ct50, ct73), USA (ct9, ct14, ct25, ct29, ct37, ct48) | Marine Mammals Exploring the Oceans Pole to Pole (MEOP) data portal
[data request: 14-11-2016] | Treasure et al. (2017)
Nachtsheim et al. (2019)
Boehme et al. (2016) | https://doi.org/10.1594/PANGAEA.899619 |
| Southern elephant seals (tracking data) | | Tosh et al. (2009a)
James et al. (2012a) | Tosh et al. (2009b)
James et al. (2012b) | https://doi.org/10.1594/PANGAEA.899619 |
| Weddell seals (tracking data) | | McIntyre et al. (2013a) | McIntyre et al. (2013b) | https://doi.org/10.1594/PANGAEA.899619 |
| Crabeater seals (tracking data) | | Nachtsheim et al. (2016a) | Nachtsheim et al. (2016b) | https://doi.org/10.1594/PANGAEA.899619 |
| Pack-ice seals (aerial surveys) | Antarctic Pack Ice Seals (APIS) programme EMAGE-I to -V | Plötz et al. (2011a-e) | Southwell et al. (2012)
Gurarie et al. (2017a, b) | https://doi.org/10.1594/PANGAEA.899619 |
| Crabeater seal densities (predicted or observed) | | Bester et al. (1995, 2002)
Flores et al. (2008)
Forcada et al. (2012) | | https://doi.org/10.1594/PANGAEA.899619 |

[revised manuscript text omitted]

**Supplementary Material**

**S1 Data processing and analysis**

The workflow of data preparation and cleaning always followed the same pattern. First we downloaded the data from the respective repository or requested the data from contact persons (see details in the main text, Section 2.3.1 to 2.4.5 and Tab. 1, 2). The collected data, which were available in various file formats (e.g., *.csv, *.shp, *.dat, *.hdf), were then checked for quality, i.e. for missing data/information, duplicates and correctness of Lat/Long coordinates. Where data sets contained missing data, "empty" cells were marked with the abbreviation NA (Not Available) and were not used for the subsequent calculations. Duplicates in the data set were deleted. If Lat/Long coordinates were missing these were subsequently added from cruise reports or other sources if possible, otherwise the data entry was also deleted. In addition, Lat/Long coordinates were checked for the correct position. Obviously wrong Lat/Long coordinates (e.g. coordinates on land, in another geographical region, etc.) were corrected along the cruise reports or checked against the principal investigators of the data sets. After the data quality check a shape file was created for each data set, which was then projected onto South Pole Lambert Azimuthal Equal Area (https://spatialreference.org/ref/esri/102020/). When working with raster data sets, the default raster cell size was 6.25 km x 6.25 km (raster size of AMSR-E 89 GHz sea ice concentration maps). Finally, the data layers were cut to the size of the study site (see main text, Section 2.1). Our data processing and statistical analyses as well as the map compilation were mainly performed using the R software (Version 3.1.2, R Core Team, 2014), QGIS (Version 2.10 "Pisa", QGIS Development Team, 2015) and the ESRI`s ArcGIS desktop software suite (Version 10.2, ESRI 2013).

**S1.1 Environmental data (IBCSO data, AMSR-E sea ice maps, FESOM data)**

For our pelagic regionalisation analysis (Fig. S2) we focused on the austral summer (December to March) and used a raster cell size of 6.25 km x 6.25 km (raster size of AMSR-E 89 GHz sea ice concentration maps). For each raster cell (i) the mean of depth (IBCSO data) and depth range (i.e. difference between maximum and minimum depth), (ii) the relative number of days with ice cover ≤ 70 % (AMSR-E sea ice maps) and (iii) the mean of temperature and salinity at the sea surface and the sea bottom (FESOM data) were calculated over the respective time periods of the environmental data sets (detailed description of environmental data sources see main text, Section 2.3, and Tab. 1). The mean of depth and depth range were ln-transformed. The parameters chosen for the pelagic regionalisation analysis are major structuring components of the Weddell Sea ecosystem and are consistent with the variables used by Raymond (2014) in a cluster approach for a circum-antarctic pelagic regionalisation.

For clustering, we used *k*-means clustering (Han et al., 2011), the most widely used numerical method for partitioning abiotic and/or biotic data in a predefined number of groups (k) (ecological examples from marine realm see e.g., Legendre et al., 2002, Hewitt et al., 2004, Zharikov et al., 2005, Verfaillie et al., 2009). To estimate the optimal number of clusters we used the gap statistic of the R package cluster (Maechler et al., 2014). The first local maximum in the gap statistic was used to define the optimal number of cluster. Due to the large amount of data, the gap statistic could not be applied to the complete data matrix (119,862 samples x 7 variables). Therefore, data subsets were extracted from the complete data matrix using a permutation approach and the gap statistic were applied to each of the data subsets. Finally, the median of the data subsets with respect to the optimal number of clusters was used for *k*-means clustering.

**S1.2 Ecological data**

**S1.2.1 Zooplankton**

**Antarctic krill (adults)**

The habitat suitability model of the Antarctic krill (Fig. S3) was developed with R (R Core Team, 2014) using the biodiversity modelling package biomod2 (Thuiller et al., 2009 and 2014). Biomod is freely available and is probably the best known and most established software in the modelling world of ecologists, geographers and conservationists, combining predictive results from different models (Hao et al., 2018 and references therein).

All models were run with presence-absence data on Antarctic krill (detailed description of data sources see main text, Section 2.4.1, and Tab. 2). The predictor variables used in our final model were defined in a stepwise procedure. First, we fed biomod2 with more than 20 environmental predictors and the model was run. The relative importance of each variable was evaluated by the following permutation procedure: Once the model is calibrated, a standard prediction is generated. Then, one of the predictor variables is randomised and a new prediction is made. The Pearson's correlation coefficient (r) between that new prediction and the standard prediction is used to measure this variable's relative importance in the model (= 1 - r; for more details on the permutation procedure see Thuiller et al., 2012). Variables with low importance were then excluded from the subsequent permutation, and the relative importance in the model of each remaining variable was measured again. Based on this permutation procedure (10 permutations in total) we reduced the number of variables to the most important predictors without negatively influencing the model performance. Thus, for our final predictive model we used the following five environmental variables (ranked by decreasing mean importance value calculated by biomod2): (i) dissolved oxygen (WOA13 data), (ii) ice coverage (AMSR-E sea ice maps), (iii) temperature (FESOM data), (iv) bathymetry (IBCSO data) and (v) chlorophyll-a concentration (SeaWiFS data) (detailed description of environmental data sources see main text, Section 2.3, and Tab. 1). All data used in the models came from near the sea surface in austral summer (January to March).

In our modelling approach, we focused on nine commonly used modelling techniques, which include regression, classification and machine learning methods, as described by Elith and Graham (2009): generalised linear model (GLM), generalised boosting model, generalised additive model, classification tree analysis, artificial neural network, surface range envelope, flexible discriminant analysis, multiple adaptive regression splines, random forest). Three evaluation methods, i.e. relative operating characteristic (ROC), true skill statistic (TSS) and accuracy, were used. Each modelling technique was calibrated with 70 % of the data (random sample from the total data set) and the remaining 30 % of the data were used to evaluate their performances (Thuiller, 2003). In total 270 calibrated models (9 different models x 10 replicates of pseudo-absences x 3 evaluation runs) were used for the model synthesis where the different models were combined into a single ensemble model (EM). For the development of our EM, all models were scaled applying a binomial GLM as implemented in biomod2 to ensure comparable model results. Out of the 270 individual models we selected those models for our EM with a TSS threshold higher than 0.65 (i.e., good prediction accuracy accord to Thuiller et al., 2010). Furthermore, we ground-truthed our EM against krill catch data from CCAMLR (see main text, Section 2.4.1, and Tab. 2) by calculating the percentage of krill catches in the areas with different predicted habitat suitability (high to unsuitable).

**Antarctic krill (larvae)**

The map of the interpolated abundances of krill larvae in the WSMPA Planning Area (see Fig. S4) was done with the ArcGIS spatial analyst in the ArcGIS desktop software suite (ESRI Inc., 2011) using the inverse-distance

weighting (IDW) method, one of the most commonly used deterministic models in spatial interpolation (e.g., Lu and Wong, 2008). The interpolation was performed with log-transformed abundance data (detailed description of data sources see main text, Section 2.4.1, Tab. 2). The output cell size (x, y) was set to 1000 m and the distance coefficient power to 2. The interpolated abundances were finally expressed for a radius of 30 km around each data record.

**Ice krill**

The map of the interpolated abundances of ice krill in the WSMPA Planning Area (Fig. S5) was developed in the same way as the Antarctic krill map (see use of interpolation in previous paragraph). For a detailed description of the data sources see Section 2.4.1 and Table 2 in the main text.

The potential ice krill habitat (Fig. S6) was approximated by water depth from 0 m to 550 m (IBCSO data) and mean sea surface temperature ≤ 0°C (FESOM data) (detailed description of environmental data sources see main text, Section 2.3, Tab. 1). The biological characteristics of ice krill were taken from the Biogeographic Atlas of the Southern Ocean (Cuzin-Roudy et al., 2014).

**S1.2.2 Zoobenthos**

**Sponges**

The map of the occurrence of sponges in the WSMPA Planning Area (Fig. S7) was finally also generated using the IDW method (see use of interpolation in Section S1.2.1 "Antarctic krill (larvae)"). The previous data processing focused on the consolidation of two different data sets (one quantitative, one semi-quantitative; for detailed description of data sources see main text, Section 2.4.2, and Tab. 2). We transformed the quantitative data into the same four-category system as the semi-quantitative data (i.e. absent, rare, common, very common) by creating a Monte Carlo sample using Sobol low-discrepancy sequences to develop a Weibull distribution (n = 10,000,000). Within the Weibull distribution, the following classes were identified (i) class 0 (absent) = 0, (ii) class 1 (rare) = 0 to mean - standard deviation (SD), (iii) class 2 (common) = mean - SD to mean and (iv) class 3 (very common) = mean to mean + SD. The quantitative data were classified according to these classes and merged with the semi-quantitative data. The interpolated data were finally expressed for a 10 nm radius around each data record according to CCAMLR Conservation Measure 22-09 (2012).

**Echinoderms**

The potential habitat for echinoderms in the WSMPA Planning Area (Fig. S8) was developed with JMP (S.A.S. Institute Inc.) using Ward`s (1963) minimum variance method, which has been widely used for calculating distances between clusters since its first description (examples from marine realm see e.g., Verfaillie et al., 2009, Weise et al., 2010, Neukermans et al., 2016). A cluster analysis with a species x station matrix was performed for Asterioidea, Ophiuroidea and Holothuroidea respectively (detailed description of data sources see main text, Section 2.4.2, and Tab. 2). All species occurred only in two stations or less were excluded from the clustering. The results of the cluster analyses were then linked to various environmental data sets. Water temperature best reflected the occurrence of a particular echinoderm community, and therefore their habitat was approximated by bottom water temperature ≤ -1° (FESOM data).

**S1.2.3 Fish**

**Antarctic silverfish (larvae and adults)**

The map of the interpolated abundances of Antarctic silverfish in the WSMPA Planning Area (Fig. S9) was developed using the IDW method (see use of interpolation in Section S1.2.1 "Antarctic krill (larvae)"). For a detailed description of the data sources see Section 2.4.1 and Table 2 in the main text. The interpolated data were finally expressed for a 10 nm radius around each data record according to CCAMLR Conservation Measure 22-09 (2012).

**Demersal fish**

The habitat suitability model of demersal fish (Fig. S10) was developed with R (R Core Team, 2014) using the biodiversity modelling package biomod2 (Thuiller et al., 2009 and 2014).

All models were run with presence-absence data on demersal fish (detailed description of data sources see main text, Section 2.4.3, and Tab. 2). The predictor variables that were used for our final predictive model were defined in the same iterative process as described under "Antarctic krill (adults)". The environmental variables finally used for our modelling approach were (with decreasing variable importance): (i) distance to coast, (ii) bathymetry, (iii) calcium carbonate, (iv) broad benthic positioning index, (v) silica, (vi) dissolved oxygen, (vii) biogenic silica, (viii) total organic carbon, (ix) nitrate, (x) salinity, (xi) temperature, (xii) current velocity, (xiii) slope and (xiv) phosphate (detailed description of environmental data sources see Section 2.3 and Tab. 1 in main text). All data used in the models came from the sea bottom in austral summer (January to March). Distance to coast, i.e. the Euclidean distance to the nearest land from each raster cell centroid (cell size: 8.02 km x 8.02 km), was calculated with the GRASS GIS package *v.distance* (Soimasuo et al., 1994) in QGIS 2.10 "Pisa". The coastline derived from IBCSO Version 1.0 DBM (Arndt et al., 2013). Slope and broad scale benthic positioning index (BPI) was also derived from IBCSO and were calculated with the Benthic Terrain Modeler Version 3.0 extension for the ArcGIS desktop software suite (ESRI Inc., 2011). For the calculation of the broad scale BPI, the inner radius was set to 5 km and the outer radius to 125 km according to Jerosch et al. (2016).

In the modelling approach, we focused on the same modelling techniques as described under "Antarctic krill (adults)". ROC and TSS were used as evaluation methods. Each modelling technique was calibrated with 70 % of the data (random sample from the total data set) and the remaining 30 % of the data were used to evaluate their performances (Thuiller, 2003). A total of 135 calibrated models (9 different models x 3 replicates of pseudo-absences x 5 evaluation runs) were used for the EM synthesis where all models were scaled applying a binomial GLM as implemented in biomod2 to ensure comparable model results. Out of the 135 individual models we selected those models for our EM with a TSS threshold higher than 0.9 (i.e., high or excellent prediction accuracy accord to Thuiller et al., 2010).

**Antarctic toothfish (adults)**

The probability model of Antarctic toothfish occurrence in the WSMPA Planning Area (Fig. S11) was developed as a function of depth as recommended by the CCAMLR Working Group on Ecosystem Monitoring and Management (WG-EMM). Following analytical steps were performed:

(i)  We calculated the standard descriptive parameters of catch per unit effort (CPUE) data on the Antarctic toothfish (CCAMLR fisheries data) per depth interval i ($mCPUE_i$) with a depth interval width of 100 m

(depth interval mean depth: $0\ m \leq D_i \leq 2600\ m$, $D_{i+1} - D_i = 50\ m$). Depth intervals with less than five CPUE data points were not considered.

(ii) A Monte Carlo sample was built for each depth interval i (n = 10,000) by randomly drawn samples from a log-normal distribution with the same mean and standard deviation as the CPUE data in each depth interval.

(iii) Outliers were defined as data points below $Q1 - 3.0\ x\ IQR$ or above $Q3 + 3.0\ x\ IQR$ per depth interval i where Q1 and Q3 are the 25% and 75% quartiles, respectively, and IQR is the interquartile range, i.e. the difference between Q1 and Q3. Thus, only extreme data points, that are "far out" (Tukey 1977), were excluded from the subsequent model fit.

(iv) We fitted a 4 parameter Weibull model to the simulated median $mCPUE_i$ per depth interval i,
$mCPUE_i = if(D_i \leq x0 - b \times ((c-1)/c)\^(1/c), 0,\ a * ((c-1)/c)\^((1-c)/c) * (abs((D_i - x0)/b + ((c-1)/c)\^(1/c))\^(c-1)) * exp(-abs((D_i - x0)/b + ((c-1)/c)\^(1/c))\^c + (c-1)/c))$
The model selection based on R (R Core Team 2014) using the package fitdistrplus (Delignette-Muller et al. 2014).

(v) The median water depth (IBCSO data) was calculated for each raster cell  (cell size: 6.25 km x 6.25 km) , was assigned to the respective depth interval and the corresponding $mCPUE_i$ value - calculated by the Weibull model for this depth interval - was mapped (detailed description of IBCSO data see main text, Section 2.3, and Tab. 1). Finally, the potential habitat of the Antarctic toothfish was bounded from 550 to 2 000 m according to CCAMLR Conservations Measures and fishing practice as recommended by WG-EMM (WG-EMM-16 report, para. 3.6).

**S1.2.4 Flying and non-flying seabirds**

**Breeding and non-breeding Adélie penguins**

The probability of occurrence of breeding and non-breeding Adélie penguins during foraging (Fig. S13, S14) was developed with R (R Core Team, 2014) using the R package crawl (Johnson, 2015). The continuous-time correlated random walk model developed by Johnson et al. (2008) has become established in Antarctic science in recent years in order to estimate more accurately the locations of tracked seabirds and pinnipeds along their trajectory (see e.g., Warwick-Evans et al., 2018 and 2019; Baylis et al., 2019).

Here, we used the random walk model to generate predictions of the location of each tracked Adélie individual on an hourly time scale (detailed description of penguin tracking data see main text, Section 2.4.4, and Tab. 2). Raw ARGOS data were first processed by assigning error values to the different ARGOS location quality codes, i.e. location code 3 (= highest accuracy of ARGOS position estimate) was set off against the lowest error value, the highest error was assigned to location code B (= lowest accuracy of ARGOS position estimate). Subsequently, simulated track-lines between the temporally sequenced ARGOS positions or each tracked individual, were generated by the continuous-time correlated random walk model, binned onto a 6.25 km x 6.25 km spatial grid and pooled per raster cell so that the final data layers (one for breeding, one for non-breeding Adélies) identifies the areas that were used most often by tracked Adélies. Buffer areas (i.e. a 50 km buffer and a 50-100 km ring buffer) around each colony - shown on the final map of breeding Adélie penguins - were adopted in accordance with the recommendations of the 2[nd] international workshop on the identification of CCAMLR MPAs in Planning Domain 1 (WG-EMM-15/42 and references therein).

**Breeding Emperor penguins**

The probability model of Emperor penguins occurrence during foraging in breeding season (Fig. S15) was developed as a function of distance from colony and colony size (Fretwell et al., 2012 and 2014) as well as sea ice concentration (AMSR-E sea ice maps; detailed description of data source see main text, Section 2.3, and Tab. 1).

*Analysis 1: Probability model of penguin occurrence as a function of distance from colony and of colony size*

To calculate the distances from colony for foraging, we used a raster grid with a spatial resolution of 6.25 km x 6.25 km (as for sea ice concentration). We calculated the Euclidian distance for each raster pixel centroid $j$ to each emperor penguin breeding colony $i$. Thus, the probability of occurrence $P1_{i,j}$ of one penguin from colony $i$ in centroid $j$ was calculated by the following approximation:

$$P1_{i,j} = \left(\frac{1}{\sqrt{\pi}}\right) * e^{\left(\frac{-\left(3*\frac{d_{i,j}}{d_{max}}\right)^2}{2}\right)} \tag{1}$$

where $d_{max}$ is the maximum foraging distance to breeding colony (here $d_{max} = 190$ km; derived from Zimmer et al. (2008) and reference therein by mean maximum foraging distance to the colony of male penguins in winter of 106 km (standard deviation (SD) = 28 km) plus three SD, i.e. 106 km + 3*28km = 190 km), and $d_{i,j}$ is the Euclidean distance (in km) between colony $i$ and centroid $j$, which was calculated by:

$$d_{i,j} = \left(\sqrt{\left(x_i - x_j\right)^2 + \left(y_i - y_j\right)^2}\right) - d.ice_{edge_i} \tag{2}$$

where $d.ice\_edge_i$ is the distance of colony to the shelf ice edge. Distances $d_{i,j} \leq 0$ were set to 1. Subsequently, different boundaries of ice shelf edge were adjusted by a 10 km puffer, which was subtracted from the distances $d_{i,j}$, too, and a reclassification was performed again ($d_{i,j} \leq 0$ were set to 1).

Then, the probability of penguin occurrence $P1_{i,j}$ from colony $i$ in centroid $j$ was normalised between 0 and 1 (i.e. $0 \leq P1_{i,j} \leq 1$). Finally, all $P1_{i,j}$ were added for each centroid $j$ and normalised again to a range between 0 and 1:

$$P1_j = \frac{\sum_{i=1}^{n} P1_{i,j}}{\max(\sum_{i=1}^{n} P1_{i,j})} \tag{3}$$

where $n$ is the number of emperor penguin breeding colonies.

To account for breeding colony size (number of animals), each probability of penguin occurrence $P1_{i,j}$ was weighted with the best population estimate (BE) for this emperor penguin colony according to Fretwell et al. (2012).

$$P1'_{i,j} = P1_{i,j} * BE_i \tag{4}$$

Subsequently, all $P1'_{i,j}$ were added for each centroid $j$ and normalised to a range between 0 and 1 (i.e. $0 \leq P1'_{i,j} \leq 1$):

$$P1'_j = \frac{\sum_{i=1}^{n} P1'_{i,j}}{\max(\sum_{i=1}^{n} P1'_{i,j})} \quad (5)$$

where $n$ is the number of emperor penguin breeding colonies.

*Analysis 2: Probability model of penguin occurrence as a function of sea ice concentration*

The probability model of penguin occurrence as a function of sea ice concentration was calculated in the following steps: (1) A sigmoid transfer function was applied (eq. 6) to achieve an even distribution of the mean sea ice concentration data; (2) the ice index data ($ICj$) were normalised to a range between 0 and 1 (eq. 7); and (3) the probability of penguin occurrence was calculated using the transformed data and a hyperbolic *tanh*-function (eq. 8). The mean sea ice concentration was calculated for the breeding period of emperor penguins (June to January) from 2002 to 2011.

$$IC_j = \frac{1}{1 + e^{(-\ln(x + 10^{-5}) * gain)}} \quad (6)$$

with x = mean sea ice concentration/100 and gain set to 6.23.

Subsequently, the ice index data ($ICj$) were normalised to a range between 0 and 1:

$$IC_j = std_{IC_j} = \frac{IC_j - \min(IC_{j_1}, IC_{j_2} \dots IC_{j_n})}{\max(IC_{j_1}, IC_{j_2} \dots IC_{j_n}) - \min(IC_{j_1}, IC_{j_2} \dots IC_{j_n})} \quad (7)$$

For the probability model of penguin occurrence we have assumed that the penguin preference does not relate linearly to sea ice conditions, but with a sigmoid pattern, i.e. areas with medium sea ice concentration are already suitable foraging grounds. This sigmoid pattern was modelled by the following *tanh*-function:

$$P2_j = \frac{\tanh\left(\pi * (IC_j * 2 - 1)\right) + 1}{2} \quad (8)$$

*Analysis 3: Combining the distance/colony size model with the sea ice concentration model*

An overall probability of penguin occurrence $P_j$, i.e. a combination of the distance/colony size model and the sea ice concentration model, was calculated by the following equation:

$$P_j = \frac{(P1_j * P2_j) - \min(P1_j * P2_{j_1}, P1_j * P2_{j_2} \dots P1_j * P2_{j_n})}{\max\left(P1_j * P2_{j_1}, P1_j * P2_{j_2} \dots P1_j * P2_{j_n}\right) - \min(P1_j * P2_{j_1}, P1_j * P2_{j_2} \dots P1_j * P2_{j_n})} \quad (9)$$

**Antarctic petrel**

The potential foraging habitats of the Antarctic petrel (Fig. S16) was developed as a function of (i) sea ice concentration (AMSR-E sea ice maps), (ii) bathymetry (IBCSO data) and (iii) sea water temperature (FESOM data; detailed description of data sources see main text, Section 2.3, and Tab. 1).

As preferred ice regime of the Antarctic petrel we focused on the marginal ice zone, i.e. 15 % - 80 % ice coverage, according to van Franeker (1996) and Ainley et al. (1984, 1994). Data on sea ice concentration were reclassified as first step, i.e. a value of 1 was assigned to each cell with ice cover 15 % - 80 %, whereas cells with ice cover less than 15 % and more than 80 % were set to 0. Then, for each grid cell, the relative number of days (in %) for which a given grid cell had an ice cover between 15 % and 80 % was calculated for the breeding period (January to March) from 2002 to 2011. Subsequently, eight classes regarding the frequency of occurrence of the marginal ice zone were defined and scaled between 0 and 1.

We used abundance data from Ainley and Jacobs (1981) and calculated mean Antarctic petrel densities for three depth classes, i.e. (1) deep ocean: > 2600 m, (2) continental slope and shelf break: 2600 to < 600 m and (3) continental shelf: the remainder of the continental shelf. Then, the mean densities were scaled between 0 and 1. Finally, bathymetric data (IBCSO) were used to identify the three different depth zones in the Weddell Sea Planning Area.

According to Ainley et al. (1984) Antarctic petrels seem to prefer water temperatures colder than 0.5°C. Thus, sea surface temperature (SST) data (FESOM) were reclassified for each raster cell, i.e. value $3 = SST \leq 0.5°C$ in all three months (January to March), $2 = SST \leq 0.5°C$ in only two months, $1 = SST \leq 0.5°C$ in only one month and $0 = SST > 0.5°C$ in all three months. Subsequently, the values were scaled between 0 and 1.

Finally, we approximated the potential foraging habitat of Antarctic petrel by stacking the three environmental proxies and corresponding data layers, respectively, and assigning different weighting factors to the proxies. The highest weighting factor was assigned to sea ice concentration (weighting factor: 1) as we assume sea ice as the major structuring component of the Antarctic petrel foraging habitat. Bathymetry and sea water temperature, in contrast, got lower weighting factors of 0.75 and 0.25, respectively.

Subsequently, we combined our model approach with the model results from Descamps et al. (2016) as recommended by the CCAMLR Scientific Committee (SC-CAMLR-XXXV report, paras. 5.14 - 5.28). Descamps et al. (2016) kindly provided us with the shape files showing the modelled kernel utilization summer and winter distribution of Antarctic petrel breeding at Svarthamaren.

We combined the kernel utilization distribution (hereafter kernel UD) model from Descamps et al. (2016) with our model by the following procedure:

(i)     We calculated a weighting factor $wf_i$ for each level of kernel UD (i.e. for 30, 60 and 95 % kernel UDs) by the following equation:

$$wf_i = \frac{\max(k_{UD})}{k_{UDi}} \tag{1}$$

where $\max(k_{UD})$ is 30 derived from the 30 % kernel UD, i.e. core area - high intensity of use, and $k_{UDi}$ is the respective kernel UD.

(ii)     We computed the probability of Antarctic petrel occurrence $P_i$ for each grid cell (i) by:

$$P_i = \frac{(\frac{x_{i_{AWI\_model}} + (100 * wf_{i_{Descamp\ et\ al.\_summer}}) + (100 * wf_{i_{Descamp\ et\ al.\_winter}})}{3})}{100 * \max(x_{i_{AWI_{model}}}, 100 * wf_{i_{Descamp\ et\ al.\_summer}}, 100 * wf_{i_{Descamp\ et\ al.\_winter}})} \tag{2}$$

where $x_{i_{AWI\_model}}$ is our model value (i.e. 5, 20, 35, 50 or 100).

**S1.2.5 Pinnipeds**

The probability of pinniped occurrence based on tracking data (Fig. S17) was developed with R (R Core Team, 2014) using the R package crawl (Johnson, 2015; see examples of Antarctic studies using crawl in Section S1.2.4 "Breeding and non-breeding Adélie penguins").

Here, we used the random walk model from Johnson et al. (2008) to generate 100 simulated track-lines between the temporally successive ARGOS positions for each tracking data set on pinnipeds (detailed description of pinniped tracking data see main text, Section 2.4.5, and Tab. 2). Only random track-lines were generated where the maximum speed of a pinniped between successive positions was ≤ 2.5 m s-1. The simulated track-lines were binned onto our standard spatial grid (cell size: 6.25 km x 6.25 km) and pooled per raster cell so that the final data layer identifies the areas that were used most often by tracked pinnipeds.

The map on seal densities in the WSMPA Planning Area (Fig. S18) was developed combining modelled and interpolated densities of seals. Predictive density values on crabeater seals were derived from Flores et al. (2008) and Forcada et al. (2012) and were pooled in case of areas where both studies presented model results.

Interpolated densities of seals were derived from APIS point data (unspecified taxa) and observed crabeater seal densities (see Bester et al., 1995, 2002; see detailed data description data in main text, Section 2.4.5, and Tab. 2). From APIS point data, seal densities (i.e. individuals/km²) were calculated using the count method for line transect data (e.g., Bester and Odendaal, 2000, Hedley and Buckland, 2004). We used non-standardised data for the density calculations as the APIS data set is based on video material, and thus at least observer related factors potentially influencing the probability of animal detection are not relevant to consider. The seal densities from Bester et al. (1995) were averaged over the different sampling dates for each transect, and the densities per sampling zones (inner, middle, outer zone; see Bester et al., 2002) were converted from square nautical mile to square kilometer. Finally, all transects were subdivided in sections of circa 5.5 km according to Bester et al. (2002) using QGIS 2.0 "Dufour" with the QChainage plugin and the density values of the respective transect was assigned to each section for the interpolation approach. We applied the IDW method (see also Section S1.2.1 "Antarctic krill (larvae)") with the output cell size (x, y) of 2000 m and the distance coefficient power of 2. The search radius setting, i.e. the number of points, was set to 10.

**S2 Figures**

[Figure]

**Figure S1.** Distribution of data recordings per higher taxonomic group, i.e. zooplankton (a), zoobenthos (b), fishes (c), birds (d) and pinnipeds (e), across the wider Weddell Sea region, which were compiled in the context of the WSMPA planning initiative.

[Figure]

**Figure S2.** Pelagic regionalisation of the WSMPA Planning Area.

[Figure]

**Figure S3.** Habitat suitability predictions of adult Antarctic krill (*Euphausia superba*) in the WSMPA Planning Area.

[Figure]

**Figure S4.** Interpolated abundances of Antarctic krill larvae (*Euphausia superba*) in the WSMPA Planning Area.

[Figure]

**Figure S5.** Interpolated abundances of ice krill (*Euphausia crystallorophias*) in the WSMPA Planning Area.

[Figure]

**Figure S6.** Potential habitat of ice krill (*Euphausia crystallorophias*) in the WSMPA Planning Area.

[Figure]

**Figure S7.** Interpolated occurrences of sponges in the WSMPA Planning Area.

[Figure]

**Figure S8.** Potential habitat of a special echinoderm assemblage in the WSMPA Planning Area.

[Figure]

**Figure S9.** Interpolated abundances of Antarctic silverfish (*Pleuragramma antarctica*) in the WSMPA Planning Area.

[Figure]

**Figure S10.** Habitat suitability predictions of demersal fishes in the WSMPA Planning Area.

[Figure]

**Figure S11.** Probability model for the potential habitat of Antarctic toothfish (*Dissostichus mawsoni*) in the WSMPA Planning Area.

[Figure]

**Figure S12.** Nesting sites of demersal fish observed in the WSMPA Planning Area.

[Figure]

**Figure S13.** Modelled probability of the occurrence of breeding Adélie penguins (*Pygoscelis adeliae*) during foraging in the WSMPA Planning Area.

[Figure]

**Figure S14.** Modelled probability of the occurrence of non-breeding Adélie penguins (*Pygoscelis adeliae*) during foraging in the WSMPA Planning Area.

[Figure]

**Figure S15.** Modelled probability of the occurrence of Emperor penguins (*Aptenodytes forsteri*) during foraging in breeding season.

[Figure]

**Figure S16.** Probability model for the potential habitat of Antarctic petrel (*Thalassoica antarctica*) in the WSMPA Planning Area.

[Figure]

**Figure S17.** Modelled probability of seal occurrence in the WSMPA Planning Area.

[Figure]

**Figure S18.** Modelled and interpolated seal abundances in the WSMPA Planning Area.

**S3 Tables**

**Table S2.** Detailed list of adult Antarctic krill (*Euphausia superba*) data with survey name, station number per survey and respective source of data, which were used from the database KRILLBASE within the WSMPA planning initiative.

| Survey name | Station | Data source |
|---|---|---|
| bas1985sib | 4a and 5 | UK data (SIBEX cruise) |
| epa1993saf | 211-216 | South African data |
| epa1995saf | 1-14 | South African data |
| epa1998saf | 27 | South African data |
| epa1995bon | 189 | South African data |
| epa1996xxx | 5 and 6 | South African data |
| epa1989ikt | 7, 8, 10, 11 | Soviet data |
| epa1990mkx | 149-173 | Soviet data |
| epa1989smt | 5, 6, 13, 14 | Soviet data |
| epa1989ikt | 7, 8, 10, 11 | Soviet data |
| his1928dis-top | 46 | UK historical data |
| his1931wsc-top | 548-552 | UK historical data |
| his1932dis-str | 813a, 815a, 816a, 822a | UK historical data |
| his1932dis-top | 814, 818-820 | UK historical data |
| his1933dis-str | 1148a, 1150a, 1151a, 1153a | UK historical data |
| his1933dis-top | 1149, 1152 | UK historical data |
| his1935dis-str | 1519a | UK historical data |
| his1935dis-top | 1513-1528 | UK historical data |
| his1935wsc-top | 892 and 893 | UK historical data |
| his1937dis-str | 2004a | UK historical data |
| his1937dis-top | 1998-2000, 2002, 2003, 2006-2012 | UK historical data |
| his1937dis-ver | 2010a, 2012a | UK historical data |
| his1939dis-str | 2596a, 2598a, 2600a, 2606a-2608a, 2010a | UK historical data |
| his1939dis-top | 2543-2563, 2595, 2599, 2601, 2603, 2604, 2609 | UK historical data |
| his1939dis-ver | 2543a, 2545a, 2547a | UK historical data |
| pol1984wit | 165 and 168 | Polish data; Witek et al. (1985) |
| vsi2004lak | 1021, 1023-1025, 1028-1031, 1033-1037, 100-1042, 1044-1046, 1048-1050, 1053-1056, 1059-1060, 1062, 1065, 1066, 1068-1071, 1075-1078, 1080, 1084, 1086, 1087, 1090-1092, 1095-1097, 1100-1102, 1116-1117 | German data (LAKRIS cruise 2004) |
| vsi1980pol | 20, 21, 23, 24, 27-29, 31, 33-35, 51, 52, 54-79, 82-85, 87-101, 105-111 | German data (MV Polarsirkel cruise 1979/80) |
| vsi1981pol | 89, 91, 94, 97-99, 101, 106, 108, 110, 111, 126-128, 130, 132, 134, 136, 138, 143, 178, 179, 185, 186, 191, 193, 195, 197, 203, 205, 208-211, 213-215, 217, 219, 224, 226-235, 237, 282 | German data (MV Polarsirkel cruise 1980/81) |
| vsi1976xxx | 161, 162, 176, 183, 185, 186, 189, 190 | German data (RV Walther Herwig cruise 1975/76) |
| vsi1978xxx | 330-337, 339, 340a, 341b, 342c, 343d, 344e, 345 | German data (RV Walther Herwig cruise 1977/78) |

**S4 References**

Ainley, D. G., Jacobs, S. S.: Seabird affinities for ocean and ice boundaries in the Antarctic. Deep-Sea Res. PT I, 28, 1173-1185, https://doi.org/10.1016/0198-0149(81)90054-6, 1981.

Ainley, D. G., O'Connor, E. F., and Boekelheide, R. J.: The marine ecology of birds in the Ross Sea, Antarctica. Ornithol. Monogr., 32, 97 pp., https://doi.org/10.2307/40166773, 1984.

Ainley, D. G., Ribic, C. A., and Fraser, W. R.: Ecological structure among migrant and resident seabirds of the Scotia-Weddell Confluence region. J. Anim. Ecol., 63, 347-364, https://doi.org/10.2307/5553, 1994.

Arndt, J. E., Schenke, H. W., Jakobsson, M., Nitsche, F.-O., Buys, G., Goleby, B., Rebesco, M., Bohoyo, F., Hong, J., Black, J., Greku, R., Udintsev, G., Barrios, F., Reynoso-Peralta, W., Morishita, T., and Wigley, R.: The International Bathymetric Chart of the Southern Ocean (IBCSO) Version 1.0 - A new bathymetric compilation covering circum- Antarctic waters, Geophys. Res. Lett., 40, 3111 –3117, https://doi.org/10.1002/grl.50413, 2013.

Baylis, A. M. M., Tierney, M., Orben, R. A., Warwick-Evans, V., Wakefield, E., Grecian, W. J., Trathan, P., Reisinger, R., Ratcliffe, N., Croxall, J., Campioni, L., Catry, P., Crofts, S., Dee Boersma, P., Galimberti, F., Granadeiro, J. P., Handley, J., Hayes, S., Hedd, A., Masello, J. F., Montevecchi, W. A., Pütz, K., Quillfeldt, P., Rebstock, G. A., Sanvito, S., Staniland, I. J., and Brickle P.: Important at-sea areas of colonial breeding marine predators on the Southern Patagonian Shelf, Sci. Rep., 9**,** 8517, https://doi.org/10.1038/s41598-019-44695-1, 2019.

Bester, M. N. and Odendaal, P. N.: Abundance and distribution of Antarctic pack ice seals in the Weddell Sea, in: Antarctic Ecosystems: Models for Wider Ecological Understanding, edited by Davison, W., Howard-Williams, C., and Broady, P., Caxton Press, Christchurch, 51-55, 2000.

Bester, M. N., Erickson, A. W., and Ferguson, J. W. H.: Seasonal change in the distribution and density of seals in the pack ice off Princess Martha Coast, Antarctica, Antarct. Sci., 7, 357-364, https://doi.org/10.1017/S0954102095000502, 1995.

Bester, M. N., Ferguson, J. W. H., and Jonker, F. C.: Population densities of pack ice seals in the Lasarev Sea, Antarctica, Antarct. Sci., 14, 123-127, https://doi.org/10.1017/S0954102002000676, 2002.

CCAMLR Conservation Measure 22-09: Protection of registered vulnerable marine ecosystems in subareas, divisions, small-scale research units, or management areas open to bottom fishing, available at: https://www.ccamlr.org/en/measure-22-09-2012, 1p., 2012.

Cuzin-Roudy, J., Irisson, J.-O., Penot, F., Kawaguchi, S., and Vallet, C.: Southern Ocean Euphausiids, in: Biogeographic Atlas of the Southern Ocean, edited by: De Broyer, C., Koubbi, P., Griffiths, H. J., Raymond, B., d'Udekem d'Acoz, C., Van de Putte, A., Danis, B., David, B., Grant, S., Gutt, J., Held, C. Hosie, G., Huettmann, F., Post, A., and Ropert-Coudert, Y., Scientific Committee on Antarctic Research, Cambridge, 309-320, 2014.

Delignette-Muller, M. L., Dutang, C., Pouillot, R., and Denis, J.-B.: Fitdistrplus: help to fit of a parametric distribution to non-censored or censored data, R package version 1.0-2, https://cran.r-project.org/src/contrib/Archive/fitdistrplus/, 2014.

Descamps, S., Tarroux, A., Cherel, Y., Delord, K., Godø, O. R., Kato, A., Krafft, B. A., Lorentsen, S.-H., Ropert-Coudert, Y., Skaret, G., Varpe, Ø.: At-sea distribution and prey selection of Antarctic petrels and commercial krill fisheries. PLoS ONE, 11, e0156968, doi:10.1371/journal.pone.0156968, 2016.

Elith, J. and Graham, C. H.: Do they? How do they? WHY do they differ? On finding reasons for differing performances of species distribution models, Ecography, 32, 66-77, https://doi.org/10.1111/j.1600-0587.2008.05505.x, 2009.

Environmental Systems Research Institute (ESRI): ArcGIS, Release 10.2, Redlands, CA, 2013.

Flores, H., Haas C., van Franeker, J. A., and Meesters, E.: Density of pack-ice seals and penguins in the western Weddell Sea in relation to ice thickness and ocean depth, Deep-Sea Res. PT. II, 55, 1068-1074, https://doi.org/10.1016/j.dsr2.2007.12.024, 2008.

Forcada, J., Trathan, P. N., Boveng, P. L., Boyd, I. L., Burns, J. M., Costa, D. P., Fedak, M., Rogers, T. L., and Southwell, C. J.: Responses of Antarctic pack-ice seals to environmental change and increasing krill fishing, Biol. Cons., 149, 40-50, https://doi.org/10.1016/j.biocon.2012.02.002, 2012.

Fretwell, P. T., LaRue, M. A., Morin, P., Kooyman, G. L., Wienecke, B., Ratcliffe, N., Fox, A. J., Fleming, A. H., Porter, C., and Trathan, P. N.: An emperor penguin population estimate: the first global, synoptic survey of a species from space, PLoS ONE, 7(4), e33751, https://doi.org/10.1371/journal.pone.0033751, 2012.

Fretwell, P. T., Trathan P. N., Wienecke, B., and Kooyman, G. L.: Emperor penguins breeding on iceshelves, PLoS ONE, 9, e85285, https://doi: 10.1371/journal.pone.0085285, 2014.

Han, J., Kamber, M., and Pei, J.: Cluster analysis: Basic concepts and methods, in: Data mining: concepts and techniques, edited by: Han, J., Kamber, M., and Pei, J., Morgan Kaufmann, Waltham, MA, USA, 443-494, 2011.

Hao, T., Elith, J., Guillera-Arroita, G., and Lahoz-Monfort, J. J.: A review of evidence about use and performance of species distribution modelling ensembles like BIOMOD, Divers. Distrib., 25, 839-852, https://doi.org/10.1111/ddi.12892, 2019.

Hedley, S. L., and Buckland, S. T.: Spatial models for line transect sampling. J. Agr. Biol. Envir. St., 9, 181-199, https://doi.org/10.1198/1085711043578, 2004.

Hewitt, J. E., Thrush, S. F., Legendre, P., Funnell, G. A., Ellis, J., and Morrison, M.: Mapping of marine soft-sediment communities: integrated sampling for ecological interpretation, Ecol. Appl., 14, 1203-1216, https://doi.org/10.1890/03-5177, 2004.

Jerosch, K., Kuhn, G., Krajnik, I., Scharf, F., and Dorschel, B.: A geomorphological seabed classification for the Weddell Sea, Antarctica, Mar. Geophys. Res., 37, 127-141, https://doi.org/10.1007/s11001-015-9256-x, 2016.

Johnson, D. S., London, J. M., Lea, M.-A., and Durban, J. W.: Continuous-time correlated random walk model for animal telemetry data, Ecology, 89, 1208-1215, https://doi.org/10.1890/07-1032.1, 2008.

Johnson, D.: Crawl: fit continuous-time correlated random walk models to animal movement data, R package version 1.5, https://cran.r-project.org/src/contrib/Archive/crawl/, 2015.

Legendre, P., Ellingsen, K. E., Bjørnbom, E., and Casgrain, P.: Acoustic seabed classification: improved statistical method, Can. J. Fish. Aquat. Sci., 1085-1089, https://doi.org/10.1139/f02-096, 2002.

Lu, G.Y., and Wong, D. W.:An adaptive inverse-distance weighting spatial interpolation technique, Comput. Geosci.-UK, 34, 1044-1055, https://doi.org/10.1016/j.cageo.2007.07.010, 2008.

Maechler, M., Rousseeuw, P., Struyf, A., Hubert, M., Hornik, K., Studer, M., and Roudier, P.: Cluster: cluster analysis extended Rousseeuw et al., R package version 1.15.2., https://cran.r-project.org/src/contrib/Archive/cluster/, 2014.

Neukermans, G., Reynolds, R. A., and Stramski, D.: Optical classification and characterization of marine particle assemblages within the western Arctic Ocean, Limnol Oceanogr, 61, 1472-1494, https://doi.org/10.1002/lno.10316, 2016.

QGIS Development Team: QGIS Geographic Information System. Open Source Geospatial Foundation Project. http://qgis.osgeo.org, 2015.

R Core Team: R: A language and environment for statistical computing, R Foundation for Statistical Computing, Vienna, Austria, http://www.R-project.org/, 2014.

Raymond, B.: Pelagic Regionalisation, in: Biogeographic Atlas of the Southern Ocean, edited by: De Broyer, C., Koubbi, P., Griffiths, H. J., Raymond, B., d'Udekem d'Acoz, C., Van de Putte, A., Danis, B., David, B., Grant, S., Gutt, J., Held, C. Hosie, G., Huettmann, F., Post, A., and Ropert-Coudert, Y., Scientific Committee on Antarctic Research, Cambridge, 418-421, 2014.

SC-CAMLR-XXXV: Domain 3 and 4 - Weddell Sea, in: Report of the thirty-fifth meeting of the Scientific Committee, Hobart, Australia, 17 - 21 October 2016, available at: https://www.ccamlr.org/en/sc-camlr-xxxv, 51-54, 2016.

Soimasuo, J., Neteler, M., Blazek, R., Landa, M., Metz, M., and Cho, H.: GRASS GIS package v.distance, http://grass.osgeo.org/grass70/manuals/v.distance.html, 1994.

Thuiller, W.: BIOMOD - optimizing predictions of species distributions and projecting potential future shifts under global change, Glob. Change Biol., 9, 1353-1362, https://doi.org/10.1046/j.1365-2486.2003.00666.x, 2003

Thuiller, W., Lafourcade, B., Engler, R., and Araújo, M. B.: BIOMOD - A platform for ensemble forecasting of species distributions,. Ecography, 32, 369-373, https://doi.org/10.1111/j.1600-0587.2008.05742.x, 2009.

Thuiller, W., Lafourcade, B., and Araújo, M.: Presentation Manual for BIOMOD, available at: https://pdfs.semanticscholar.org/5522/2d3a8501206807a9a0a32a2b7e62260f53af.pdf, 35 pp., 2010.

Thuiller, W., Georges, D., and Engler, R.: Biomod2: Ensemble platform for species distribution modeling, R package version 3.1-64, https://cran.r-project.org/src/contrib/Archive/biomod2/, 2014.

Tukey, J. W.: Exploratory data analysis., Addison-Wesley Publishing Company, Reading, MA, USA, 1977.

Van Franeker, J. A., Gavrilo, M., Mehlum, F., Veit, R. R., and Woehler, E. J.: Distribution and abundance of the Antarctic Petrel, Waterbirds, 22, 14-28, https:/doi.org/10.2307/1521989, 1999.

Verfaillie, E., Degraer, S., Schelfaut, K., Willems, W., and Van Lanckerad, V.: A protocol for classifying ecologically relevant marine zones, a statistical approach, Estuar. Coast. Shelf Sci., 83, 175-185, https://doi.org/10.1016/j.ecss.2009.03.003, 2009.

Ward, J. H.: Hierarchical grouping to optimize an objective function, J. Amer. Statist. Ass., 58, 236-244, https://doi.org/10.1080/01621459.1963.10500845, 1963.

Warwick-Evans, V., Ratcliffe, N., Lowther, A. D., Manco, F., Ireland, L., Clewlow, H. L., and Trathan, P. N.: Using habitat models for chinstrap penguins Pygoscelis antarctica to advise krill fisheries management during the penguin breeding season, Divers. Distrib., 24, 1756-1771, https://doi.org/10.1111/ddi.12817, 2018.

Warwick-Evans, V., Downie, R., Santos, M., and Trathan, P. N.: Habitat preferences of Adélie *Pygoscelis adeliae* and Chinstrap Penguins *Pygoscelis antarctica* during pre-moult in the Weddell Sea (Southern Ocean), Polar Biol., 42, 703-714, https://doi.org/1007/s00300-019-02465-9, 2019.

Weise, M. J., Harvey, J. T., and Costa, D. P.: The role of body size in individual-based foraging strategies of a top marine predator, Ecology, 91, 1004-1015, https://doi.org/10.1890/08-1554.1, 2010.

WG-EMM-15/42: Report of the Second International Workshop for identifying Marine Protected Areas (MPAs) in Domain 1 of CCAMLR (Palacio San Martín, Buenos Aires, Argentina, 25 to 29 May 2015), 42 pp., 2015.

WG-EMM-16: MPA Planning Domains 3 and 4 - Weddell Sea, in: Report of the Working Group on Ecosystem Monitoring and Management, Bologna, Italy, 4 - 15 July 2016, available at: https://www.ccamlr.org/node/88723, 252-254, 2016.

Witek, Z., Kittel, W., Czykieta, H., Żmijewska, M. I., and Presler, E.: Macrozooplankton in the southern part of Drake Passage and in the Bransfield Strait during BIOMASS-SIBEX (December 1983 - January 1984), Pol. Polar Res., 6, 95-115, 1985.

Zharikov, Y., Skilleter, G. A., Loneragan, N. R., Taranto, T., and Cameron, B. E.: Mapping and characterising subtropical estuarine landscapes using aerial photography and GIS for potential application in wildlife conservation and management, Biol. Conserv., 125, 87-100, https://doi.org/10.1016/j.biocon.2005.03.016, 2005.

Zimmer, I., Wilson, R.P., Gilbert, C., Beaulieu, M., Ancel, A., and Plötz, J.: Foraging movements of emperor penguins at Pointe Géologie, Antarctica, Polar Biol., 31, 229-243, https://doi.org/10.1007/s00300-007-0352-5, 2008.

---

## Author Response (AR3)

**Reply to comments from the editor**

Dear Mr Fleischer,

We uploaded the final version of our paper including the extended table with the availability of the raw data and the supplement for data processing instructions.

Kind regards,

Katharina Teschke and co-authors